# PQMASS: PROBABILISTIC ASSESSMENT OF THE QUALITY OF GENERATIVE MODELS USING PROBABILITY MASS ESTIMATION

**Pablo Lemos** [*1,2,3,4,5†]   **Sammy Sharief** [*1,2,3†]   **Nikolay Malkin**[1,2,3,6]   **Salma Salhi**[1,2,3,7]
**Connor Stone**[1,2,3]   **Laurence Perreault-Levasseur**[1,2,3,5,7]   **Yashar Hezaveh**[1,2,3,5,7]

[1]Mila – Québec Artificial Intelligence Institute   [2]Ciela Institute   [3]Université de Montréal
[4]SanboxAQ   [5]CCA – Flatiron Institute   [6]University of Edinburgh   [7]Trottier Space Institute
[†] These authors contributed equally.

## ABSTRACT

We propose a likelihood-free method for comparing two distributions given samples from each, with the goal of assessing the quality of generative models. The proposed approach, PQMass, provides a statistically rigorous method for assessing the performance of a single generative model or the comparison of multiple competing models. PQMass divides the sample space into non-overlapping regions and applies chi-squared tests to the number of data samples that fall within each region, giving a $p$-value that measures the probability that the bin counts derived from two sets of samples are drawn from the same multinomial distribution. PQMass does not depend on assumptions regarding the density of the true distribution, nor does it rely on training or fitting any auxiliary models. We evaluate PQMass on data of various modalities and dimensions, demonstrating its effectiveness in assessing the quality, novelty, and diversity of generated samples. We further show that PQMass scales well to moderately high-dimensional data and thus obviates the need for feature extraction in practical applications. Code: https://github.com/Ciela-Institute/PQM.

## 1 INTRODUCTION

Generative modeling – the task of inferring a distribution given a set of samples – is an important and ubiquitous task in machine learning. Generative machine learning has witnessed the development of a succession of methods for distribution approximation in high-dimensional spaces, including variational autoencoders (VAEs, Kingma & Welling, 2014), generative adversarial networks (GANs, Goodfellow et al., 2014), normalizing flows (Rezende & Mohamed, 2015), and score-based (diffusion) generative models (Ho et al., 2020). With advancements in generative models, evaluating their performance using rigorous, clearly defined metrics and criteria has become increasingly essential.

The ability to distinguish between true and modeled distributions has become increasingly crucial, particularly in the context of AI safety across various applications. This is evident in concerns regarding data privacy (*e.g.*, Somepalli et al., 2023; Hitaj et al., 2017) and the proliferation of AI-generated content across different domains of the internet (*e.g.*, Knott et al., 2024). As AI systems become more pervasive, accurately assessing the fidelity of generated distributions to their real-world counterparts is essential for ensuring the integrity and safety of AI applications.

In scientific applications, recent years have seen an increased application of deep generative models. For example, they have been used as expressive plug-and-play priors (*e.g.*, Song et al., 2022; Chung & Ye, 2022; Rozet & Louppe, 2023; Kawar et al., 2022; Dou & Song, 2024; Chung et al., 2023; Feng et al., 2023; Dia et al., 2023; Drozdova et al., 2024; Xue et al., 2023; Remy et al., 2023; Flöss et al., 2024; Feng et al., 2024), noise models for explicit likelihood inference (*e.g.*, Legin et al., 2023; Adam et al., 2023), and density estimators in conjunction with deep emulators for simulation-based inference (*e.g.*, Cranmer et al., 2020; Rezende & Mohamed, 2015; Papamakarios et al., 2021; Price et al., 2018; Greenberg et al., 2019; Papamakarios et al., 2019). However, the adoption of

[*]Correspondence: `Pablo.lemos@sandboxquantum.com` and `sammy.sharief@umontreal.ca`

Figure 1: Illustration of the PQMass statistic. **Left:** Two sets of samples $(x_i)_{i=1}^{n} \sim p$, and $(y_i)_{i=1}^{m} \sim q$, are shown as blue and red points. For a given region $R \subseteq X$, the fraction of samples in $R$ for each set follows a binomial distribution. We can test the hypothesis that the two binomial distributions are the same. **Right:** Partitioning of the space into non-overlapping regions $\{R_i\}_{i=1}^{n_R}$, *e.g.*, a Voronoi tessellation, the distribution of samples in the regions follows a multinomial distribution. PQMass performs hypothesis tests on equality of the two multinomial distributions' parameters to obtain a p-value.

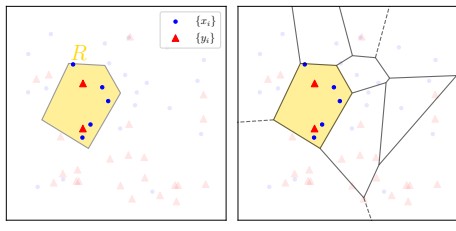

these methods in scientific data analysis demands rigorous accuracy in uncertainty quantification. Specifically, it is crucial to verify that these generative models accurately represent the full distribution of their training data.

When evaluating generative models, one is interested in three qualitative properties (Stein et al., 2023; Jiralerspong et al., 2023): **Fidelity** refers to the quality and realism of outputs generated by a model, *i.e.*, how indistinguishable generated samples are from real data. **Diversity** pertains to the range and variety of outputs a model can produce. For example, a model that misses a mode in the training data has lower diversity. **Novelty** refers to the ability of a model to generate new, previously unseen samples that are not replicas of the training data yet are coherent and meaningful within the context of the task. The trade-offs between these properties are complex and depend on the function class (*e.g.*, model architecture) used for the generative model, the learning objective, properties of the data distribution, etc. (Stein et al., 2024). For example, samples from a model that simply copies the training data will exhibit high fidelity and diversity but will lack novelty; on the other hand, an excessively smooth model will generate novel, diverse samples but will lack fidelity.

Two classes of methods for comparing generative models to ground truth distributions exist: **sample-based methods**, which compare generated samples to true samples, and **likelihood-based methods**, which make use of the likelihood of the data under the model. It has been observed that likelihood-based methods, such as negative log-likelihood (NLL) of test data, show too much variance to be useful in practice and are 'saturated' on standard benchmarks, *i.e.*, they do not correlate well with sample fidelity, particularly in high-dimensional settings Theis et al. (2016); Nalisnick et al. (2019); Nowozin et al. (2016); Yazici et al. (2020); Le Lan & Dinh (2021). On the other hand, most sample-based methods cannot measure the combination of fidelity, diversity, and novelty. For example, the Fréchet Inception Distance (FID; Heusel et al., 2017) and the Inception Score (IS; Salimans et al., 2016; Sajjadi et al., 2018) measure fidelity and diversity, but not novelty, while precision and recall (Salimans et al., 2018) measure only fidelity and diversity, respectively; the authenticity score (Alaa et al., 2022) measures only novelty. Recently, the Feature Likelihood Divergence (FLD; Jiralerspong et al., 2023) was proposed as a measure that captures fidelity, novelty, and diversity. However, FLD essentially relies on the approximation of the underlying distribution by a Gaussian mixture model and consequently requires feature extraction and compression in order to function in high-dimensional settings.

In this work, we propose **PQMass**, a statistical framework for evaluating the quality of generative models that measures the combination of fidelity, diversity, and novelty and scales well to moderately high dimensions, without the need of dimensionality reduction. PQMass allows us to accurately estimate the relative probability of a generative model given a test dataset by examining the *probability mass* of the data over appropriately chosen regions. The main idea motivating PQMass is that given a measurable subset of the sample space, the number of data points lying in the subset follows a binomial distribution with a parameter equal to the region's (unknown) true probability mass. By extension, given a partition of the space into regions, the counts of points falling into the regions follow a multinomial distribution. PQMass carefully defines partitions of the space into regions defined by Voronoi cells, quantizes sets of samples from two distributions into these cells, and uses two-sample tests on parameters of the resulting multinomial distributions. This approach provides a p-value that estimates the quality of a generative model (the probability of a model given the training data). It also allows the comparison of various competing models by analyzing the ratio of their probabilities.

PQMass satisfies the following desiderata:

(1) Since PQMass works with the *integrals* of the true and estimated probability density functions over regions, it does not make any assumptions about the true underlying density field, *e.g.*, its smoothness or approximability by simple families.

(2) It is statistically principled, allowing us to directly upper-bound the answer to the question: 'What is the probability that two sets of samples were drawn from the same distribution?'

(3) PQMass is computationally efficient and scales well with the dimensionality of the data, allowing us to directly evaluate generative models of high-dimensional data, such as images. However, PQMass can also readily work in feature space if desired.

(4) PQMass is general and flexible, allowing the evaluation of generative models on any type of data where a metric can be defined, including images, tabular data, and time series.

## 2 THEORY AND METHODS

Our problem statement is the following: We have two collections of i.i.d. samples $(x_i)_{i=1}^m$, $x_i \sim p$ and $(y_i)_{i=1}^n$, $y_i \sim q$, where both $p$ and $q$ are probability distributions[1] on $\mathbb{R}^d$. We are interested in testing the statistical hypothesis that $p = q$, that is, the two sets of samples came from the same distribution.

To achieve this, we will develop tests on well-understood distributions derived from $p$ and $q$ that would upper-bound the probability that $p = q$ and show that they almost surely distinguish two nonequal distributions as the number of samples increases.

### 2.1 TESTING EQUALITY OF QUANTIZED DISTRIBUTIONS USING SAMPLES

**Equality of distributions from equality of probabilities of regions.** We begin with some elementary facts from probability and real analysis. Recall that two probability measures $p$ and $q$ are equal if they assign the same mass to all measurable sets, *i.e.*,

$$p(R) = q(R) \qquad \forall R \subseteq X, R \text{ measurable.} \tag{1}$$

Because $p$ and $q$ are Borel measures, it is equivalent to verify (1) only for open sets $R$, or a smaller generating set, such as all polyhedra or products of intervals (see, *e.g.*, Rudin, 1987).

Let $\mathbb{R}^d = R_1 \sqcup R_2 \sqcup \cdots \sqcup R_{n_R}$ be a partition of $\mathbb{R}^d$ into disjoint measurable regions. Define $\pi : \mathbb{R}^d \to \{1, 2, \ldots, n_R\}$ as the function that assigns to each point $x \in \mathbb{R}^d$ the index of the region it belongs to, *i.e.*, $x \in R_{\pi(x)}$. If $p$ is a distribution on $\mathbb{R}^d$, then the pushforward measure $\pi_* p$ is a distribution on $\{1, 2, \ldots, n_R\}$, with mass function $\pi_* p(j) = p(R_j)$. This distribution is a quantized version of $p$, where samples are replaced by the index of the region into which they fall.

With this definition, (1) can be restated: $p = q$ if and only if $\pi_* p = \pi_* q$ for *all* partitions $\mathbb{R}^d = R_1 \sqcup R_2$, or, more generally, $\mathbb{R}^d = R_1 \sqcup \cdots \sqcup R_{n_R}$ (note that $\pi$ implicitly depends on the partition). As we will return to in §2.2, it is sometimes sufficient to consider restricted classes of partitions.

The main idea behind PQMass is that the probability that $p = q$ is upper-bounded by the probability that $\pi_* p = \pi_* q$ for a given partition. Next, we will consider how to test the hypothesis that $\pi_* p = \pi_* q$ using samples from $p$ and $q$ given a fixed partition. In §2.2 we discuss the choice of partition and propose an effective method.

**Estimating probability masses of regions by sampling.** Fix a partition $\mathbb{R}^d = R_1 \sqcup R_2 \sqcup \cdots \sqcup R_{n_R}$ and define $\pi$ as above. If $x \sim p$ is a random variable with distribution $p$, then $\pi(x)$ is a categorical variable with distribution $\pi_* p$, *i.e.*, taking the value $j \in \{1, 2, \ldots, n_R\}$ with probability $p(R_j)$.

Suppose now that $x_1, \ldots, x_m \sim p$ are i.i.d. samples from $p$. We then have $\pi_*(x_i) \sim \pi_* p$ for all $i$. Defining $k\left((x_i)_{i=1}^m, R\right) = \sum_{i=1}^m \mathbb{1}[x_i \in R]$, where $\mathbb{1}$ is the indicator function, as the number of samples in a region $R$, we have that $k\left((x_i)_{i=1}^m, R_j\right)$ is a Binomial$(m, p(R_j))$ random variable. In particular, the *proportion* of samples in $R_j$ is an unbiased and strongly consistent estimator of $p(R_j)$, since the mean of $k\left((x_i)_{i=1}^m, R_j\right)/m$ is $p(R_j)$ and the law of large numbers implies that this estimate converges to $p(R_j)$ almost surely as $n \to \infty$.

Generalizing, we have that the vector of counts in all the regions follows a multinomial distribution:

$$\mathbf{k}\left((x_i)_{i=1}^m\right) := \left(k\left((x_i)_{i=1}^m, R_1\right), \ldots, k\left((x_i)_{i=1}^m, R_{n_R}\right)\right) \sim \text{Multinomial}\left(m, \left(p(R_1), \ldots, p(R_{n_R})\right)\right), \tag{2}$$

and the vector of empirical proportions of samples in the regions is an unbiased and strongly consistent estimator of the vector of true probabilities $(p(R_1), \ldots, p(R_{n_R}))$.

---

[1]We assume $p$ and $q$ are Borel measures, *i.e.*, all open sets are measurable. This is true if they are absolutely continuous with respect to Lebesgue measure or to the induced measure on an embedded submanifold.

**Testing the equality of multinomial distributions.** Suppose that $(x_i)_{i=1}^m$, $x_i \sim p$ and $(y_i)_{i=1}^n$, $y_i \sim q$ are i.i.d. samples from $p$ and $q$ respectively. According to (2), the vectors of counts $\mathbf{k}\left((x_i)_{i=1}^m\right)$ and $\mathbf{k}\left((y_i)_{i=1}^n\right)$ follow multinomial distributions with parameters $\left(m, (p(R_1), \ldots, p(R_{n_R}))\right)$ and $\left(n, (q(R_1), \ldots, q(R_{n_R}))\right)$, respectively. Testing the statistical hypothesis that $\pi_* p = \pi_* q$ is the same as testing the hypothesis that the vectors $(p(R_1), \ldots, p(R_{n_R}))$ and $(q(R_1), \ldots, q(R_{n_R}))$ are equal. Thus the problem is reduced to testing the equality of two multinomial distributions given samples from each, which is a standard problem in statistics. We discuss how to approach this problem from a frequentist perspective.

In a frequentist approach, there are multiple ways to measure whether two multinomial distributions are the same (see, *e.g.*, Anderson et al., 1974; Zelterman, 1987; Plunkett & Park, 2019; Bastian et al., 2024). In this paper, we use the Pearson $\chi^2$ test (Rao, 1948; 2002).

We start by defining the probability that a point randomly chosen from among the $m + n$ points $(x_i)_{i=1}^m$, $(y_i)_{i=1}^n$ falls in region $R_j$:

$$\hat{p}_j := \frac{k\left((x_i)_{i=1}^m, R_j\right) + k\left((y_i)_{i=1}^n, R_j\right)}{m + n}.$$

If the $m + n$ points were randomly partitioned into two subsets of sizes $m$ and $n$, then the expected number of points in $R_j$ within the two subsets would be

$$\hat{N}_j^{(1)} := m\hat{p}_j, \qquad \hat{N}_j^{(2)} := n\hat{p}_j, \tag{3}$$

respectively. This motivates the definition of the $\chi^2_{\text{PQM}}$ statistic using the definition of the Pearson chi-squared:

$$\chi^2_{\text{PQM}} := \sum_{j=1}^{n_R} \left[ \frac{\left(k\left((x_i)_{i=1}^m, R_j\right) - \hat{N}_j^{(1)}\right)^2}{\hat{N}_j^{(1)}} + \frac{\left(k\left((y_i)_{i=1}^n, R_j\right) - \hat{N}_j^{(2)}\right)^2}{\hat{N}_j^{(2)}} \right]. \tag{4}$$

It is known that, given *any* distributions, this statistic follows a $\chi^2$ distribution with $n_R - 1$ degrees of freedom. Therefore, we can calculate the p-value of the test as

$$\text{p-value}(\chi^2_{\text{PQM}}) \equiv \int_{\chi^2_{\text{PQM}}}^{+\infty} \chi^2_{n_R - 1}(z)\, dz. \tag{5}$$

This p-value provides an estimate of the probability that the $\pi_* p = \pi_* q$, *i.e.*, that the distributions quantized into the regions $R_j$ coincide.

## 2.2 CHOOSING AN EFFECTIVE PARTITION

The test described above depends on the choice of partition, with some choices of regions giving more powerful test than others. For example, (5) is unlikely to give meaningful p-values if nearly all the mass of both $p$ and $q$ is captured by one of the regions. Similarly, one could be 'unlucky' in the choice of regions and obtain that $\pi_* p$ happens to be very close to $\pi_* q$, making the test uninformative with a small number of points. Thus, one key element in PQMass is the choice of the regions $R_j$.

**Voronoi cells.** Inspired by the *Tests of Accuracy with Random Points* (TARP; Lemos et al., 2023a) framework for estimating posterior coverage, we propose to define the regions as Voronoi cells.

Recall that given a set of reference points $z_1, \ldots, z_{n_R}$, the Voronoi cell corresponding to $z_j$ is defined as the set of points that is closer to $z_j$ than to any other $z_{j'}$. To be precise, we can break ties according to the index, yielding the following definition:

$$x \in R_j \iff \forall j', \left[ \|x - z_j\| < \|x - x_{j'}\| \text{ or } \left[ \|x - z_j\| = \|x - x_{j'}\| \text{ and } j < j' \right] \right]. \tag{6}$$

The sets $R_1, \ldots, R_{n_R}$ then partition $\mathbb{R}^d$ into nonoverlapping regions, and the function $\pi$ that maps points to the index of the containing region is defined by $\pi(x) = \arg\min_j \|x - z_j\|$, breaking ties in favour of the lower index (see Fig. 1, right).

While the definition (6) uses the $L^2$ distance metric, one can define Voronoi cells using any metric $D : \mathbb{R}^d \times \mathbb{R}^d \to \mathbb{R}$. In Table 3 we compare this choice to that of using the $L^1$ metric, as well as with the choice of using a metric defined in a feature space.

**Choice of centres.** It remains to specify how to choose the reference points $z_j$. To ensure that the regions approximately uniformly cover both distributions, we choose to sample the reference points from a uniform mixture of the empirical distributions of samples from $p$ and $q$.

For the validity of the test in §2.1, the choice of the regions should be independent of the points used to perform the test. Thus one can use a small subset of the available sample points as reference points to construct a tessellation, then evaluate the $\chi^2_{\text{PQM}}$ statistic using the remaining points.

**Algorithm.** The full algorithmic instantiation of PQMass, given sets of points $(x_i)_{i=1}^m$ and $(y_i)_{i=1}^n$, a choice of the number of regions $n_R > 1$, and a distance metric $D$, can be summarized as follows:

1. **Define reference points**: Set $z_1, \ldots, z_{n_R}$ to a random sample from $\left\lfloor \frac{n_R}{2} \right\rfloor$ points from $(x_i)_{i=1}^m$ and $\left\lceil \frac{n_R}{2} \right\rceil$ points from $(y_i)_{i=1}^n$. Remove the choice points from the collections $(x_i)$ and $(y_i)$.
2. **Count points in Voronoi cells**: Obtain the vectors of counts $\mathbf{k}\left((x_i)_{i=1}^m\right)$ and $\mathbf{k}\left((y_i)_{i=1}^n\right)$ with respect to the regions defined by reference points $z_j$, by computing bincounts of $\left(\arg\min_j D(x_i, z_j)\right)_{i=1}^m$ and $\left(\arg\min_j D(y_i, z_j)\right)_{i=1}^n$ respectively.
3. **Test to compare multinomials**: Compute the $\chi^2_{\text{PQM}}$ and p-value via (4) and (5), respectively.

In practice, to obtain a useful metric, one can perform the test multiple times, with different sets of reference points, to reduce the variance resulting from the choice of regions.

### 2.3 CONSISTENCY GUARANTEES: WHAT INFORMATION IS LOST BY PQMASS?

We present results showing that the PQMass test is capable of distinguishing distinct distributions.

The first proposition shows that even with two regions, PQMass has nonzero probability of distinguishing two distributions.

**Proposition 2.1.** *Suppose that $p, q, r$ are Lebesgue-absolutely continuous distributions on $\mathbb{R}^d$ with $p \neq q$, that $p$ and $q$ have smooth densities, and that $r$ has full support. Then, for $n_R = 2$ and references points $z_1, z_2 \sim r$, the probability that $\pi_* p \neq \pi_* q$ is strictly positive and hence the PQMass test is consistent as $m, n \to \infty$ with $m/n$ being bounded both above and below by strictly positive, finite scalars.*

*Proof.* As $m, n \to \infty$, the bounds on $m/n$ guarantee that $m$ and $n$ grow at rates such that Fisher's test for multinomial distributions with $m$ and $n$ samples from the two distributions distinguishes them. Because $r$ has full support, the distribution of the hyperplane separating the two Voronoi cells defined with respect to samples from $r$ has positive density in the space of hyperplanes. Therefore, if the test statistic were almost surely zero, then the measures of almost all half-spaces under $p$ and $q$ would coincide. By disintegration, this would imply that the marginals of $p$ and $q$ coincide on almost all hyperplanes, *i.e.*, the Radon transforms of $p$ and $q$ are equal. This would imply the equality of $p$ and $q$, a contradiction. □

Since $\pi$ depends on $r$, the probability referred to in Prop. 2.1 is over the choice of $n_R$ samples from $r$. Of course, for "bad" choices of reference points, we may be unlucky and not distinguish $p$ and $q$ (if the two resulting cells have the same mass under $p$ and $q$). The proposition states that with positive probability, this does not happen. Similarly, as the number of *references points* $n_R$ grows to infinity, the probability that the test distinguishes two distributions approaches one. This can be understood as a consequence of the fact that the Voronoi cells 'tessellate' the space more finely as $n_R$ grows.

**Proposition 2.2.** *Let $p, q, r$ be as in the previous proposition. As $n_R \to \infty$, the probability that $\pi_* p \neq \pi_* q$ with respect to the choice of reference points $z_1, \ldots, z_{n_R} \sim r$ approaches 1.*

*Proof.* We abuse notation and use $p$ and $q$ interchangeably with their densities. Select any $x_0$ such that $p(x_0) < q(x_0)$, which exists because $p \neq q$ and both $p$ and $q$ integrate to 1 over $\mathbb{R}^d$. By continuity, there exists $\delta > 0$ such that if $\|x - x_0\| < \delta$, then $p(x) < q(x)$.

Let $B$ be the open ball of radius $\delta$ about $x_0$. By Theorem 3.1 of Gibbs & Chen (2020), the probability that the diameter of the Voronoi cell (defined using reference points sampled i.i.d. from the full-support distribution $r$) containing $x_0$ less than $\delta$ approaches 1 as $n \to \infty$. Any polytope of diameter less than $\delta$ and containing $x_0$ is contained in $B$; thus, the probability that at least one region $R_j$ is

Figure 2: Null test. We generate two sets of samples from the same Gaussian mixture model in 100 dimensions. We then measure the $\chi^2_{\text{PQM}}$ value of our test ($n_R = 100$) and repeat this process $2^{14}$ times, resampling every time. We can see that the $\chi^2_{\text{PQM}}$ value is distributed as a chi-squared distribution with $n_R - 1$ degrees of freedom, as expected.

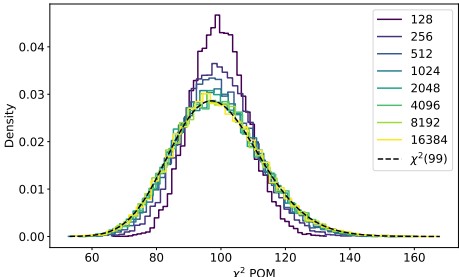

Table 1: Comparison of various sampling methods on a mixture model in $D = 2$ and a funnel in $D = 10$. The results for PQMass are compared to the Wasserstein distance, the MMD with an RBF kernel, and the Jensen-Shannon divergence.

| Model | Mixture Model ($D = 2$) | | | | Funnel ($D = 10$) | | |
| --- | --- | --- | --- | --- | --- | --- | --- |
| | $\chi^2_{\text{PQM}}$ ($n_R = 100$) | $\mathcal{W}_2$ | RBF MMD | JSD | $\chi^2_{\text{PQM}}$ ($n_R = 100$) | $\mathcal{W}_2$ | RBF MMD |
| MCMC | $1621.06 \pm 82.40$ | 8.94 | 0.0151 | 0.42 | $855.40 \pm 49.60$ | 17.41 | 0.0566 |
| FAB | $678.22 \pm 88.43$ | 5.77 | 0.0038 | 0.39 | $96.69 \pm 13.52$ | 22.08 | 0.0032 |
| GGNS | $109.32 \pm 12.36$ | 3.20 | 0.0011 | 0.35 | $472.99 \pm 18.00$ | 32.20 | 0.0358 |

contained in $B$ approaches 1 as $n \to \infty$. Such a cell would have higher mass under $q$ than under $p$, which implies the result. $\qquad\square$

Note that the $n_R \to \infty$ limit in Prop. 2.2 does not refer to performing the test with "$n_R = \infty$" to compare two $\chi^2$ distributions with infinite degrees of freedom. The proposition states that the probability that the tessellation defined using $n_R$ random reference points sampled from $r$ separates $p$ and $q$ – which is a well-defined quantity for any $n_R$ – approaches 1 as $n_R \to \infty$. These results show that, at least for 'well-behaved' densities, no information is lost by the PQMass statistic. By this, we mean that if two distributions are distinct, then the test has a positive probability of distinguishing them (and the probability approaches 1 in the asymptotic limit). This contrasts with tests that approximate the distributions by a certain fixed class (*e.g.*, tests considering only summary statistics such as the second-moment statistics used by FID). These results can likely be generalized further, which we leave for future work.

Since Prop. 2.2 is about the sensitivity of the test in the asymptotic limit, the proposition simply tells us that the sensitivity of the test approaches 1 as the number of samples grows, but does not tell us about the rate of convergence. The usefulness of the test itself is demonstrated by the empirical evidence in the experiments presented in the next section.

## 3 EXPERIMENTS

### 3.1 NULL TEST

We start by validating PQMass on a null test, by comparing two sets of samples that are known to be equivalent. As shown in Fig. 2, we use a Gaussian mixture model in 100 dimensions with 20 components as our generative model. We then repeat the following process $2^{14}$ times: We generate a number of samples (see legend) from the Gaussian mixture model and then measure the $\chi^2_{\text{PQM}}$ value of our test, with $n_R = 100$. The $\chi^2_{\text{PQM}}$ value is approximately distributed as a chi-squared distribution with $n_R - 1$ degrees of freedom, with the approximation getting better as the number of samples gets larger, as expected from §2.1.

This test, while simple, is important, as it shows that the proposed method is not biased towards rejecting the null hypothesis. We show further null tests for complex distributions in §A, and explore approximations that can be used when a limited number of samples are available in §I.

### 3.2 VALIDATION OF SAMPLING METHODS

In this section, we show how PQMass can be used to study the performance of sampling algorithms. Other sample-based metrics commonly used to evaluate sampling methods include the Wasserstein distance and maximum mean discrepancy (MMD, Gretton et al., 2012). While these sample-based

Figure 3: Various sample-based metrics as a function of the dimensionality of the data. We use a mixture of 10 equally weighted Gaussians with varying numbers of dimensions. For each, we calculate the value of both metrics when comparing samples from the same distribution (blue) and from different distributions where one of them is missing one mode (orange), repeating the test various times. Sample-based metrics should detect that a mode has been dropped.

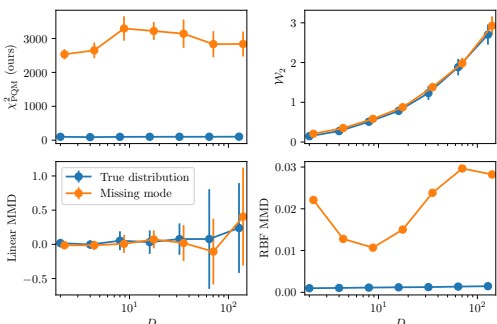

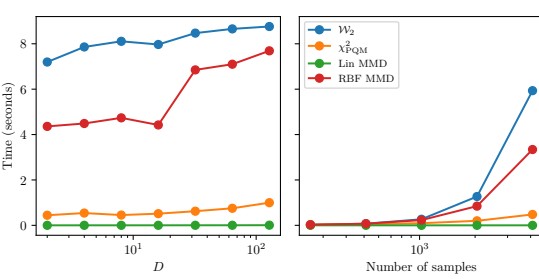

Figure 4: Computational cost of various sample-based metrics vs data dimensionality. We show the time required to calculate the value of each metric for the experiment in Fig. 3, both as a function of the dimensionality of the data (left) and the number of samples (right).

metrics are well-adapted to low-dimensional settings, they do not scale well to high-dimensional settings, as they may be sensitive to outliers or to the choice of kernels.

**PQMass is consistent with baseline metrics on simple tasks.** First, we compare PQMass to other sample-based metrics on a low-dimensional task where traditional sample-based metrics are known to perform well. We use the two-dimensional Gaussian mixture model used in Midgley et al. (2023), which has been used for benchmarking various sampling methods. We generate samples using three representative methods: flow-annealed importance sampling bootstrap (FAB, Midgley et al., 2023) a Markov chain Monte Carlo (Metropolis et al., 1953) algorithm as implemented in `emcee` (Foreman-Mackey et al., 2013)[2], and gradient-guided nested sampling (GGNS, Skilling, 2006; Lemos et al., 2023b). For each method, we generate $10,000$ samples and compare them with $10,000$ samples from the true distribution. We calculate the 2-Wasserstein distance, the mean maximum discrepancy with a radial basis function (RBF) kernel, and the Jensen-Shannon divergence (after performing a step of kernel density estimation); and compare them to the $\chi^2_{\text{PQM}}$ value with $n_R = 100$. For PQMass, we repeat this process 20 times, resampling the reference points, and report the standard deviation. We show the results in Table 1. PQMass is in good agreement, and the $\chi^2_{\text{PQM}}$ values correlate well with the other sample-based metrics, with further comparisons in §H.

**Scaling to more complex distributions.** We then repeat this experiment on a higher-dimensional task: Neal's 10-dimensional funnel distribution (Neal, 2003), another common sampling benchmark. PQMass is again in good agreement with MMD, but not with the Wasserstein distance. We show samples from each model in §C, which show that, visually, PQMass correctly identifies the best-performing sampling methods.

**PQMass detects mode-dropping in high dimension.** Finally, we study the scaling of PQMass with the dimensionality of the data. We use a uniform mixture of 10 Gaussians in $\mathbb{R}^d$, for varying $d$. For each $d$, we generate $5,000$ samples and compare them with $5,000$ samples from the same distribution. We calculate the $\chi^2_{\text{PQM}}$ value of our test, with $n_R = 100$, as well as other sample-based metrics. We then repeat the experiment, but comparing the true distribution with the distribution *with one mode dropped*. We show the results in Fig. 3. While $\mathcal{W}_2$ and linear MMD get increasingly noisier as the dimensionality increases, PQMass remains stable. Furthermore, the computational cost of radial basis MMD scales far worse with dimensionality than PQMass (Fig. 4). MMD with a linear

---

[2]Note that MCMC does not produce independent samples, as samples within the samples are correlated. While we randomly subsample 0.01 of the total number of samples in each chain, there could still be spurious correlations between these, affecting, the MCMC results. However, we find the MCMC results to be significantly worse than competing methods in these experiments, far beyond the possible effects of spurius correlations.

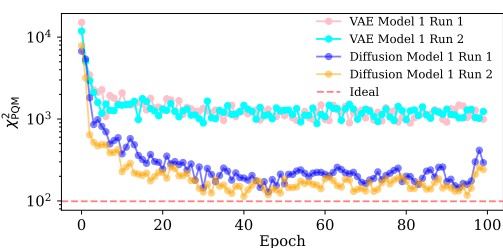

Figure 5: Tracking training progress of generative models on MNIST. After each training epoch, we generate samples and compute the $\chi^2_{\text{PQM}}$ value to assess their quality against test samples. We perform this experiment twice for each model, and see that the values agree even though unique samples were generated for each run. The diffusion model can achieve values closer to ideal ($\chi^2_{\text{PQM}}$ equal to the number of regions $n_R - 1$, here 99). $\chi^2_{\text{PQM}}$ was evaluated by retessellating and resampling the data sets.

Figure 6: Correlation of human error rate in identifying real CIFAR-10 and FFHQ images with various sample-based metrics (results of prior methods taken from Jiralerspong et al. (2023)). Each point corresponds to a single model. The result for PQMass is the $\chi^2_{\text{PQM}}$ value averaged over 20 samples. PQMass estimates the fidelity and diversity of generative models well despite not relying on a feature representation, unlike FLD and FID.

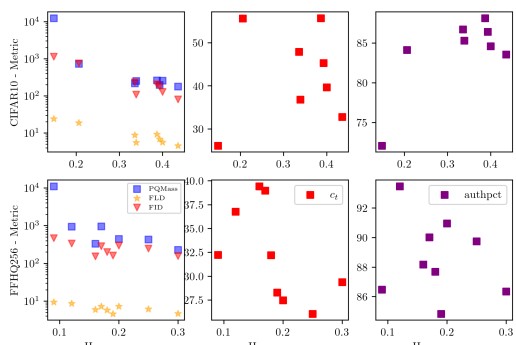

kernel is the cheapest computationally, but, as shown in Fig. 3, it is too noisy to be usable even in simple high-dimensional problems. We show results with larger number of modes dropped in §F.1.

## 3.3 ASSESSING IMAGE GENERATIVE MODELS

**Tracking training progress on small images.** In this section, we track the value of $\chi^2_{\text{PQM}}$ between model-generated and ground truth samples as we train two generative models, a variational autoencoder (VAE; Kingma & Welling, 2014) and a denoising diffusion model (Ho et al., 2020; Song et al., 2021) (see §D for model details). We train on the MNIST[3] train set and compare generated samples with the MNIST test set. After each epoch of training, we generate 2000 samples and compare them with 2000 test samples by computing $\chi^2_{\text{PQM}}$ with $n_R = 100$. We execute this experiment over the first 100 training epochs to highlight the early stages of training for the models and repeat the experiment twice. As shown in Fig. 5, as training progresses, the metric stabilizes, with fluctuations due to the stochasticity of training and the unique samples generated at each epoch.

As shown in §E.1, the VAEs fail to capture correct structures of the images, while the diffusion-generated images are more realistic. This is reflected in the low $\chi^2_{\text{PQM}}$ values for the diffusion model, which plateau at around $\chi^2_{\text{PQM}}$ closer to $\approx 99$, the ideal value.

We remark that this test was performed directly in pixel space, without dimensionality reduction or feature extraction. Thus, at least for simple images, PQMass can serve as a userful metric for tracking the quality of samples from a generative model over the course of training.

**Measuring mode coverage.** To verify that our method can measure the diversity of samples from generative models, we retrain the variational autoencoder and diffusion model described above on subsets of MNIST where all images of $N$ randomly chosen classes are missing, for varying $N$. The results, averaged over random runs and samples of classes, are shown in Fig. 7. As expected, the value of $\chi^2_{\text{PQM}}$ increases as we drop more classes for both models. In addition, we verify the ability of PQMass to detect the presence of a small mode in one of the sets in §F.2.

**PQMass correlates with human judgments.** Next, we consider the assessment of *pretrained* models on larger image datasets. We perform an experiment inspired by Stein et al. (2023); Jiralerspong et al. (2023), focusing on the CIFAR-10[4] and FFHQ256 (Karras et al., 2019) datasets. Following Stein et al. (2023), we first use the human error rate as a measure of the fidelity of the generative models.

---

[3] http://yann.lecun.com/exdb/mnist/

[4] https://www.cs.toronto.edu/~kriz/cifar.html

Figure 7: We show the results for PQMass validating generative models trained on MNIST on the Y-axis, and the number of dropped modes on the X-axis, for a variational autoencoder in blue, and a diffusion model in red. The $\chi^2_{\text{PQM}}$ value increases as we drop more modes, as expected. For this experiment, $\chi^2_{\text{PQM}}$ was evaluated by retessellating and resampling the data sets.

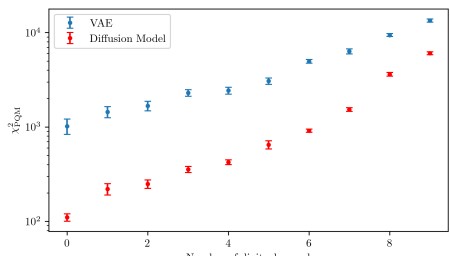

Table 2: We compare ImageNet train data, noised with Gaussian noise of different variances, to ImageNet test data. We also ran FLD for the same samples using the DinoV2 feature extractor. For PQMass, these experiments were done by retessellating and resampling the sets for evaluating $\chi^2_{\text{PQM}}$. As compared to FLD, PQMass is more sensitive to the noise despite not using a feature extractor.

| Noise $\sigma^2$ | 0.00 | 0.01 | 0.02 | 0.03 | 0.04 | 0.05 | 0.06 | 0.07 | 0.08 | 0.09 | 0.10 | 0.15 | 0.20 | 0.50 | 1.00 |
|---|---|---|---|---|---|---|---|---|---|---|---|---|---|---|---|
| $\chi^2_{\text{PQM}}$ Avg | 95.55 | 107.63 | 116.63 | 131.55 | 151.67 | 181.00 | 212.25 | 227.44 | 265.38 | 300.99 | 352.49 | 766.00 | 1252.45 | 2818.56 | 4495.07 |
| FLD | 9.29 | 8.40 | 9.15 | 9.27 | 9.53 | 10.16 | 10.54 | 10.87 | 10.96 | 11.34 | 11.93 | 15.84 | 21.79 | 42.78 | 42.54 |

We compare generated samples from each of the models to the test data for both datasets. We repeat the comparison 20 times, each time varying the reference points. We show the results in Fig. 6. We find that our chi-squared values visibly correlate with the human error rate, indicating that PQMass can effectively measure the fidelity and diversity of generative models.

We note again that these experiments were performed in pixel space. One key strength of PQMass is its scaling to high-dimensional problems: methods such as FID and FLD fail in high dimensions without the use of a feature extractor to reduce the dimensionality of the data, while PQMass continues to give meaningful results, at a much lower computational cost and without introducing potential biases through feature extraction. This ability is important because pretrained feature extractors do not exist for many modalities of data that appear in important scientific applications.

However, PQMass can also work in feature space: Table 3 shows a comparison of PQMass applied in pixel space and using InceptionV3 features (Szegedy et al., 2016).

**PQMass effectively detects small noise.** Next, we consider the detection of additive noise on ImageNet (Deng et al., 2009). We first run PQMass on a subset of the training data with added Gaussian noise of different variance, comparing it to the test data (see §E.2 for visualization). We repeat the experiment by replacing PQMass with FLD using the DinoV2 (Oquab et al., 2024) feature extractor. The results of both experiments are shown in Table 2. PQMass is more sensitive to noise, especially in the high noise regime, demonstrating that PQMass not only scales well with dimension (see §F.5 for further demonstrations of the scalability of PQMass to high-dimensional settings), but also provides a measure of distribution shift caused by increasing noise added to the data.

**Using PQMass to measure novelty and memorization.** While PQMass can measure the fidelity and diversity of model-generated samples, it does not directly measure their novelty. One approach for detecting memorization or overfitting would be to follow Jiralerspong et al. (2023) in looking at the *generalization gap* for PQMass, *i.e.*, how this metric differs between comparing the generated samples to the training data and comparing the generated samples to the validation data. However, we propose a way to detect overfitting from $\chi^2_{\text{PQM}}$ on the training data directly. Indeed, we know that a value that is too low is indicative of overfitting. For a large number of regions $n_R$ (*i.e.*, $n_R - 1$ degrees of freedom), we know the chi-square distribution is approximately Gaussian (Fig. 2). Therefore, we can use the *asymmetry* of this distribution to get a p-value that penalizes suspiciously low values of $\chi^2_{\text{PQM}}$, by considering the mirror reflection of the chi-square value around the maximum:

$$\text{p-value}_{\text{overfit}}(\chi^2_{\text{PQM}}) \equiv \int_{-\infty}^{2n_R - \chi^2_{\text{PQM}}} \chi^2_{n_R-1}(z)dz. \tag{7}$$

To study the effectiveness of this metric, we repeat the 'copycat' experiment of (Jiralerspong et al., 2023): We generate samples using one of the well-performing CIFAR-10 models used above (PFGMPP, Xu et al., 2023) and repeat our test, replacing varying fractions of the generated samples with samples from the training data. We then calculate $\chi^2_{\text{PQM}}$, first comparing the generated samples to the training data and then to the validation data.

Figure 8: Novelty. PQMass (Y-axis) when a fraction of the generated samples are replaced by samples taken from the training set (X-axis). We repeat the comparison 20 times, each time changing the reference points, and report the standard deviation. We show the value of $\chi^2_{\mathrm{PQM}}$ in the top left panel, for the test (red) and train (blue), and the p-value for the training data on the top right. We see that, as memorization increases, the gap between the train and test $\chi^2_{\mathrm{PQM}}$ increases, and the p-value goes down. The bottom pannels compare p-value$_{\mathrm{overfit}}(\chi^2_{\mathrm{PQM}})$ with the $C_T$ score and the percentage of authentic samples authpct, two metrics thats measure novelty.

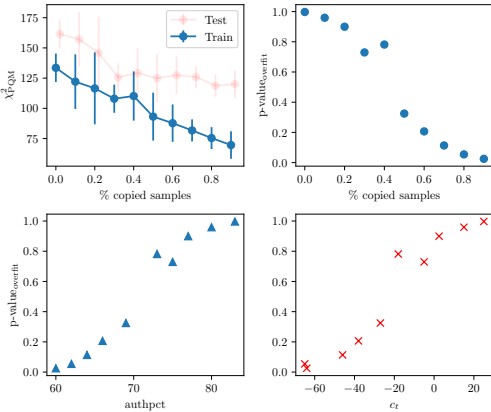

We show the results for in Fig. 8. The top left panel shows that the gap between $\chi^2_{\mathrm{PQM}}$ when comparing to the training data and when comparing to the validation data increases with the number of samples we replace. The top right panel shows the two-sided p-value, which decreases as the number of copied samples increases. These results show that both generalization gap and two-sided p-value can be used to detect memorization in generative models. The bottom two panels show the effectiveness of the p-value in (7) in capturing memorization and its agreement with an established metric, the authenticity score authpct (Alaa et al., 2022).

**Experiments in the Appendix demonstrate the effectiveness of PQMass for other data modalities, including higher-dimensional images from scientific applications (§F.5), time series (§F.3), tabular data (§F.4) and protein sequences (§F.6).**

## 4 LIMITATIONS

While PQMass performs well on various tasks, it is important to consider its limitations.

PQMass was shown to perform well with large number of samples. The statistics, however, will be too noisy for practical use if this number is too limited. Similarly, the number of regions is important: while our ablation study (§B) shows that the results are robust to various choices, in the limits $n_R = 1$ and $n_R \to N$ (where $N$ is the number of samples), sensitivity will be limited.

Second, PQMass could fail for a fixed choice of reference points defining a tessellation that fails to discriminate two distributions well. For this reason, repeating the experiment for various randomly chosen tessellations is recommended, as this effectively marginalizes the choice of tessellation (as we have done in this work). Fortunately, this is feasible, as the computational cost of PQMass is low.

Third, PQMass does require i.i.d. samples. Therefore, if a generative model produces correlated samples, the PQMass formalism cannot be used.

Finally, PQMass is only as good as the distance metric used, as shown in §G. For some modalities (for example, natural language), it is nontrivial to select a meaningful distance metric between samples.

## 5 CONCLUSION

In this paper, we have introduced a new method for quantifying the probability that two sets of samples are drawn from the same probability distribution. PQMass is based on comparing the probability mass of the two sets of samples in a set of non-overlapping regions. It relies only on the calculation of distances between points and, therefore, can be applied to high-dimensional problems. It does not require training or fitting any models. Furthermore, it does not require any assumptions about the underlying distribution and, therefore, can be applied to any type of data as long as one has access to true samples. We have shown that PQMass can be used to evaluate sampling methods and track the performance of generative models as they train.

We have shown the performance of PQMass on a variety of synthetic tasks, as well as on comparing sampling methods, comparing generative models of images, and detecting hidden signals in time series. Given the versatility (further shown in the additional experiments in §F) and low computational cost of PQMass, it can serve as a valuable tool for evaluating the quality and performance of generative models and sampling methods.

ACKNOWLEDGMENTS

This research was made possible by a generous donation by Eric and Wendy Schmidt with the recommendation of the Schmidt Futures Foundation. This work is also supported by the Simons Collaboration on "Learning the Universe". The work is in part supported by computational resources provided by Calcul Quebec and the Digital Research Alliance of Canada. Y.H. and L.P. acknowledge support from the Canada Research Chairs Program, the National Sciences and Engineering Council of Canada through grants RGPIN-2020- 05073 and 05102, and the Fonds de recherche du Québec through grants 2022-NC-301305 and 300397. N.M. acknowledges funding from CIFAR, Genentech, Samsung, and IBM.

We thank Adam Coogan, Sebastian Wagner-Carena, Joey Bose, Gauthier Gidel, Alex Tong, Alexandra Volokhova, and Marco Jiralespong for very inspiring discussions and feedback on early versions of this work. We also thank Marco Jiralespong for sharing useful data.

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

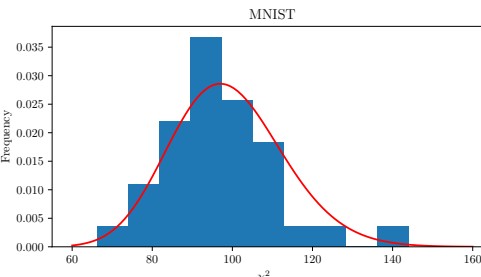 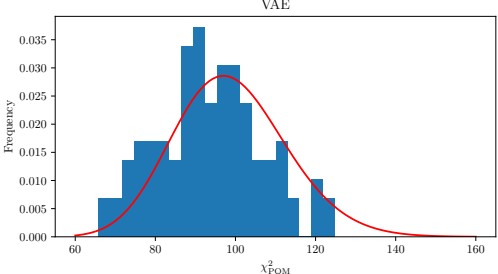

Figure 9: Null tests, similar to the one shown in Fig. 2, but for two different distributions: Samples from MNIST (left) and a VAE trained on MNIST (right). The main result of our paper holds for samples from any distribution as long as we are comparing two sets of independent samples from the same distribution. Note that, on MNIST, the fit is not perfect due to the limited number of data points.

## A  NULL TESTS

The main result of this paper, shown in §2, was shown in practice in §3.1 for a high-dimensional mixture of Gaussians. In this section, we show that it works in even more complex and high-dimensional distributions. First, we test it on the MNIST data: Using the MNIST observations themselves as samples from some underlying probability distribution. We put together the MNIST train and test data, leading to $70,000$ images. We then split them into two subsets of $35,000$ images each. To repeat the null test of §3.1, we need independent samples at every iteration (note that this is different from the error bars reported in the rest of the experiments, which just arise from repeating with random tessellations, but always the same samples). Therefore, for each iteration, we take $1,000$ images from each dataset, and perform the PQM test with $n_R = 100$, leading to 35 data points. We plot their histogram in the left panel of Fig. 9. While noisy, due to the very few data points, we see that the histogram does follow a chi-squared distribution with $n_R - 1$ degrees of freedom, as expected.

For our second experiment, we use one of the MNIST generative models described in §3.3. In this case, we can generate as many samples as we want. Because we are comparing the generative model to itself, the similarity of the generated samples to the MNIST train or test data is irrelevant for this particular test.

In practical applications, the number of samples available from real data might be limited, and in such cases, the PQMass distribution can be approximated by reusing the samples and only retessellating, which also yields good results. In practice, we have observed that, for typical sample sizes, the match between the $\chi^2_{PQM}$ distribution obtained by retessellating without resampling and the theoretical $\chi^2$ distribution is very close and more than adequate for testing the null hypothesis (we have indicated in the captions of the figures and the tables the specific experiments where this was done). However, in the cases where reusing the samples between retessellation, the retessellations are not independent since the samples are shared between each test, which might affect the obtained distribution. We explore further the impact of this approximation in §I.

## B  ABLATION STUDY

In this section, we study the effect of varying the two hyperparameters of our experiment: The number of reference points $n_R$, and the distance metric (which in all experiments in the main text is L2). We repeat the experiment on CIFAR10 described in §3.3, varying the number of reference points, as well as changing the distance metric to L1. We also show the results when using the Inception-v3 feature extractor (Szegedy et al., 2016), as done in previous work such as Jiralerspong et al. (2023). We find that the order of the models is robust to the choice of the number of reference points and the distance metric, as well as to the use of a feature extractor. Any observed differences are within the standard deviation of the experiment. We show the results in Table 3.

## C  SAMPLES FOR VALIDATION

In §3.2, we used PQMass to compare samples from various sampling algorithms to true distribution samples. Fig. 10 shows the samples from each algorithm and the underlying distribution. The

Table 3: Ablation study for CIFAR-10. We repeat the experiment of §3.3, varying the number of reference points, as well as changing the distance metric to L1. We show the results for the frequentist version of PQMass, sorted in order of decreasing $\chi^2_{\text{PQM}}$ for the default version. We find that the results are robust to the choice of the number of reference points and the distance metric. Any observed differences are within the standard deviation of the experiment.

| Model | Default ($n_R = 100$) | $n_R = 50$ | $n_R = 200$ | L1 Distance | InceptionV3 |
|---|---|---|---|---|---|
| ACGAN-Mod | $12407 \pm 10$ | $10410 \pm 808$ | $14125 \pm 995$ | $12336 \pm 944$ | $7891 \pm 1009$ |
| LOGAN | $713 \pm 72$ | $408 \pm 69$ | $1180 \pm 95$ | $728 \pm 108$ | $5750 \pm 535$ |
| BigGAN-Deep | $268 \pm 30$ | $135 \pm 16$ | $498 \pm 36$ | $238 \pm 32$ | $615 \pm 74$ |
| iDDPM-DDIM | $238 \pm 22$ | $143 \pm 17$ | $470 \pm 41$ | $244 \pm 31$ | $533 \pm 78$ |
| MHGAN | $234 \pm 39$ | $123 \pm 19$ | $470 \pm 48$ | $194 \pm 18$ | $628 \pm 64$ |
| StyleGAN-XL | $230 \pm 13$ | $119 \pm 15$ | $477 \pm 23$ | $203 \pm 23$ | $199 \pm 19$ |
| StyleGAN2-ada | $207 \pm 26$ | $128 \pm 25$ | $417 \pm 48$ | $197 \pm 23$ | $259 \pm 30$ |
| PFGMPP | $177 \pm 22$ | $108 \pm 18$ | $335 \pm 28$ | $175 \pm 16$ | $239 \pm 28$ |

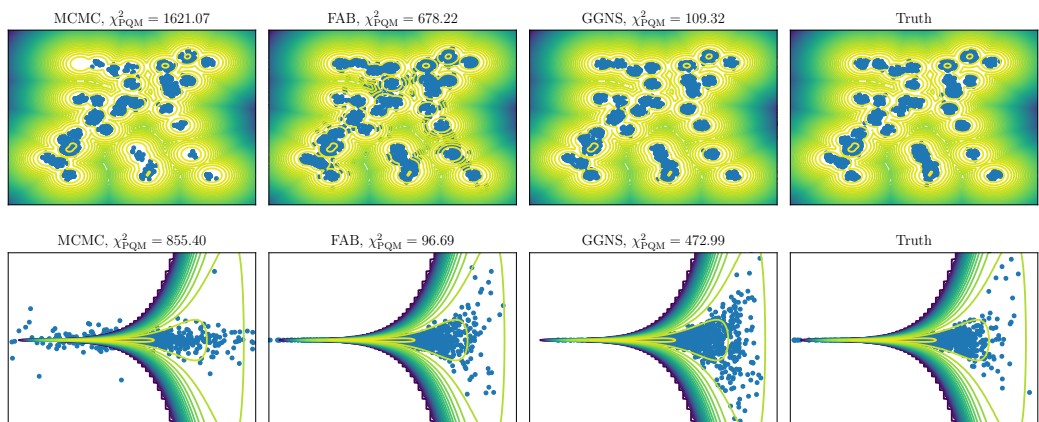

Figure 10: Samples from our the various sampling methods introduced in §3.2, and their corresponding PQMass values.

two distributions (gaussian mixture model and funnel) are often used for benchmarking sampling algorithms because of the high multimodality of the former and the complex shape of the latter. For the Gaussian mixture model (top), we see that MCMC is missing a mode, while the FAB samples are too noisy; for Neal's funnel (bottom), we see that FAB's samples look indistinguishable from true samples, while GGNS look similar but more spread out, while MCMC fails to model this distribution. In both cases, the results from $\chi^2_{\text{PQM}}$ correlate well with the similarity to true samples we can see by eye.

## D  GENERATIVE MODEL HYPERPARAMETERS

We detail the hyperparameters used for the image generative models trained in the main text. The MNIST VAE was trained with a ReLU-activated MLP encoder with layer structure $(28 \times 28) \rightarrow 512 \rightarrow 256 \rightarrow 20$, and the decoder architecture was symmetric. The model was trained using the Adam optimizer with the learning rate initially set to 0.001, decaying by a factor of 0.9 every 50 steps. For the denoising diffusion model, we used the package `score_models`[5]. Our model utilizes the NCSN++ architecture with variance-preserving noising process and is trained with batch size 256 and learning rate 0.001 (with exponential moving average decay of 0.999. Neither model deviates from standard practices and we expect similar results to hold when assessing training progress of other generative models.

---

[5]https://github.com/AlexandreAdam/torch_score_models

# E  VISUAL INSPECTION OF SAMPLES

## E.1  MNIST TRAINING PROGRESS

In the first part of §3.3, we trained four generative models, two (identical) VAEs, and two (identical) diffusion models on the MNIST dataset for 100 epochs. In Fig. 5, we show the evolution of the $\chi^2_{\text{PQM}}$ values as a training progresses. Here, in Fig. 11, we display one sample from run 1 of each generative model as a function of the training epoch to provide a qualitative understanding of the sensitivity of PQMass.

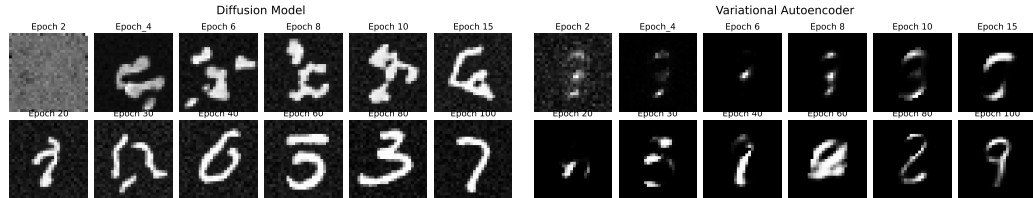

Figure 11: Samples from the run 1 diffusion model (left) as well as from the run 1 VAE model (right) as a function of training epoch.

## E.2  ADDING NOISE TO IMAGENET

In §3.3, we evaluate PQMass's and FLD's ability to detect the addition of Gaussian noise with increasing variance compared to the test set from the low to high noise regime. As shown in Table 2, PQMass is more sensitive to noise corruption than FLD. We display an example showcasing the impact of each noise level on the sample in Fig. 12 to offer an intuitive understanding of the sensitivity of each metric.

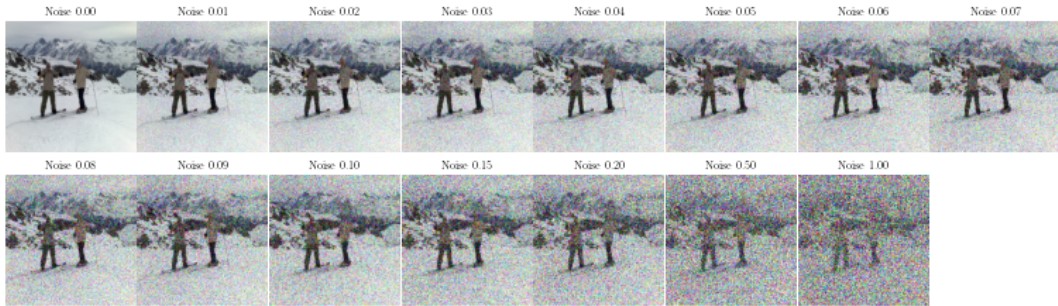

Figure 12: A sample from the ImageNet dataset shown with Gaussian noise of increasing variance added, significantly altering the image in the high noise regime.

# F  ADDITIONAL EXPERIMENTS

## F.1  GAUSSIAN MIXTURE MODEL

To evaluate our method's ability to detect the diversity of generated samples, we reuse the Gaussian Mixture Model from §3.1, but this time we systematically remove samples from $N$ modes, for varying $N$. For each $N$, we generate 5,000 samples and run our test 20 times with $n_R = 100$, using different random reference points in each trial. The results are shown in Fig. 13. As expected, the $\chi^2_{\text{PQM}}$ value increases as more modes are dropped, demonstrating that PQMass can effectively detect when a generative model fails to capture all modes, indicating reduced sample diversity.

Figure 13: Gaussian Mixture Model. We show the results for PQMass on the Y axis, and the number of dropped modes on the X axis. We repeat the comparison 20 times, each time changing the reference points, and report the standard deviation. The $\chi^2_{\text{PQM}}$ value increases as we drop more modes, as expected. For this experiment, $\chi^2_{\text{PQM}}$ was evaluated by retessellating and regenerating new data sets.

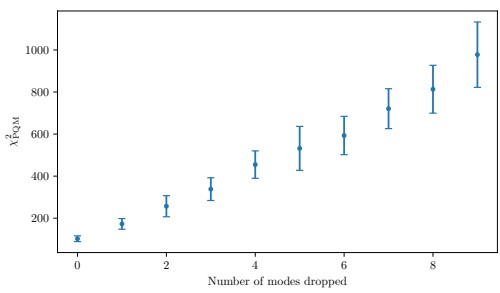

## F.2 DETECTING A SMALL MODE

We evaluate the effectiveness of PQMass at detecting a discrepant small mode between two sets of samples. To this end, we use the CIFAR-10 dataset. We select eight classes (airplanes, automobiles, birds, cats, deer, dogs, frogs, and horses) and split the dataset into two sets $X$ and $Y$, each containing 24,000 samples (3,000 from each class). To simulate a small mode, we introduce a parameter, $\alpha$, and add $3000 \cdot \alpha$ images of a new class (trucks) only to $X$, while randomly removing the same number of images from other classes to keep the sizes of $X$ and $Y$ identical. We then use PQMass as well as other existing tests to compare $X$ and $Y$.

In Table 4, we see that, as we include images from the new class, which introduces a small mode, PQMass can detect the difference between $X$ and $Y$ even for small $\alpha$. On the other hand, FLD struggles to detect the difference; FID also detects the difference (as shown by the monotonically increasing test statistic), but, unlike PQMass, does not give a point of reference allowing to compute significances (p-values).

We also should note that failing to detect that two distributions differ by a small mode as described above could affect any two-sample test by construction. One can always design a mode to be "sufficiently small" that it eludes a sample-based test, as samples from it would not appear among the test samples. It is also worth noting that in the Voronoi binning step of the PQMass test, all samples are binned, such that PQMass can be sensitive to a small missing mode even without a reference sample within said mode.

Table 4: Small mode detection. We compare the sensitivity of PQMass with FID and FLD at detecting a small mode using the CIFAR-10 dataset. The PQMass test was performed by retessellating and ressampling for every value of $\alpha$.

| $\alpha$ | PQMass | FID | FLD | $\alpha$ | PQMass | FID | FLD |
|---|---|---|---|---|---|---|---|
| 0.00 | 100.41 | 17.66 | -4.05 | 0.10 | 114.65 | 21.41 | -3.88 |
| 0.01 | 100.60 | 17.77 | -4.13 | 0.20 | 144.75 | 27.84 | -3.78 |
| 0.02 | 101.71 | 18.03 | -3.89 | 0.30 | 195.67 | 36.18 | -3.85 |
| 0.03 | 102.71 | 18.31 | -4.16 | 0.40 | 261.46 | 45.21 | -3.98 |
| 0.04 | 103.80 | 18.68 | -3.64 | 0.50 | 318.96 | 54.49 | -3.95 |
| 0.05 | 104.02 | 18.96 | -4.05 | 0.60 | 407.73 | 64.41 | -3.69 |
| 0.06 | 106.89 | 19.56 | -3.78 | 0.70 | 486.08 | 75.08 | -3.22 |
| 0.07 | 108.00 | 20.15 | -4.15 | 0.80 | 579.01 | 85.00 | -3.37 |
| 0.08 | 110.44 | 20.40 | -4.03 | 0.90 | 677.07 | 97.70 | -3.01 |
| 0.09 | 112.31 | 21.11 | -4.14 | 1.00 | 778.88 | 108.72 | -2.90 |

## F.3 TIME SERIES

For our next experiment, we show the flexibility of PQMass by applying it to a different data modality: time series data. For this, we design an experiment where we observe a noisy time series of fixed length and aim to determine whether an underlying signal is hidden within the noise. The time series

Figure 14: Time Series. We show the results for PQ-Mass on the Y-axis and the amplitude of the signal on the X-axis. We repeat the comparison 5000 times, each time generating a new time series with $A = 0$, and a new time series with $A \neq 0$. We can see that the $\chi^2_{\text{PQM}}$ value increases as $A$ grows, as expected. We also show the values of $\chi^2_{\text{PQM}}$ corresponding to the $3\sigma$ and $5\sigma$ significance levels of detection as black dashed lines.

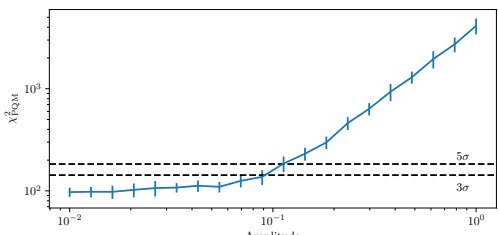

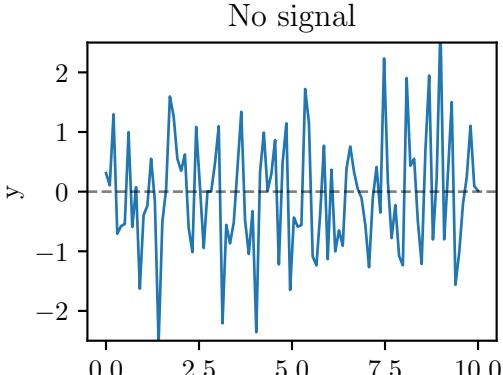 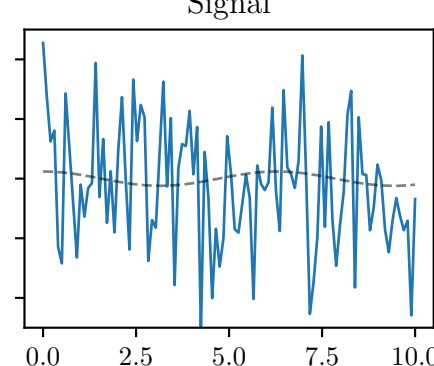

Figure 15: Examples of time series with $A = 0$ (left), and with $A \approx 0.12$ (right). PQMass can detect the signal with $5\sigma$ significance, a signal that is invisible to the naked eye (at least, to those of the authors).

is generated as follows:

$$y(t) = A \cdot \cos(t) + \eta(t), \tag{8}$$

where $A$ is the amplitude of the signal and $\eta(t)$ is i.i.d. Gaussian noise with zero mean and unit variance. If $A = 0$, then the time series is just noise. If $A \neq 0$, then there is a signal hiding in the noise. For each observation, we generate 100 data points between $t = 0$ and $t = 10$. We then repeat the following process 5000 times: We generate a time series with $A = 0$ and a time series with $A \neq 0$. We then compare the two time series using PQMass, with $n_R = 100$. We show the results in Fig. 14 for varying values of $A$. We can see that the $\chi^2_{\text{PQM}}$ value increases as $A$ grows, as expected. The plot also the values of $\chi^2_{\text{PQM}}$ corresponding to the $3\sigma$ and $5\sigma$ significance levels of detection. We see that, for this experiment, we can detect the signal with $5\sigma$ significance for $A \approx 0.12$, a signal that is invisible to the naked eye. We show an example of a time series with this amplitude, compared to one without signal in Fig. 15.

This experiment demonstrates the versatility of PQMass. Because we make no assumptions about the underlying distribution, we can apply PQMass to any type of data as long as we have access to samples. Detecting signals in noisy time series is a common problem in multiple disciplines, such as astronomy (Zackay et al., 2021; Aigrain & Foreman-Mackey, 2023), finance (Chan, 2004; Sezer et al., 2020), and anomaly detection (Ren et al., 2019; Shaukat et al., 2021). Existing methods rely on assumptions about the underlying distribution. PQMass, on the other hand, can detect that the observed signal is not consistent with samples of random noise with no assumptions on the generative process. We leave the application of PQMass to these domains for future work.

### F.4 TABULAR DATA EXPERIMENT

To show the versatility of our method on different types of data, we applied PQMass to the generation of tabular data. We used CTGAN Xu et al. (2019) as our generative model, trained on data from the Adult Census Dataset [6]. We convert categorical entries into a one-hot encoding for distance metrics

---

[6] https://archive.ics.uci.edu/dataset/2/adult

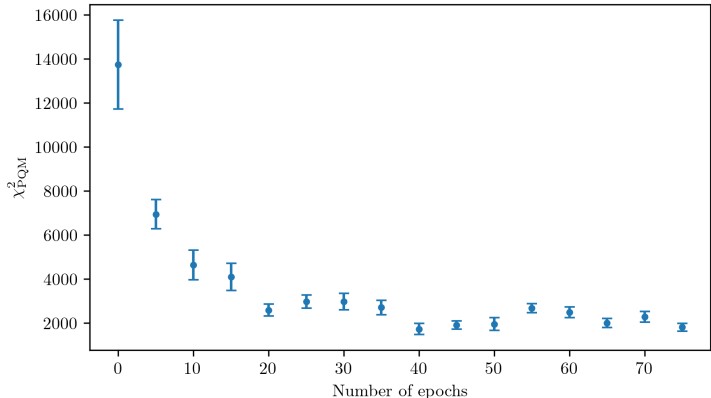

Figure 16: $\chi^2_{\text{PQM}}$ for generative model on tabular data, as training progresses.

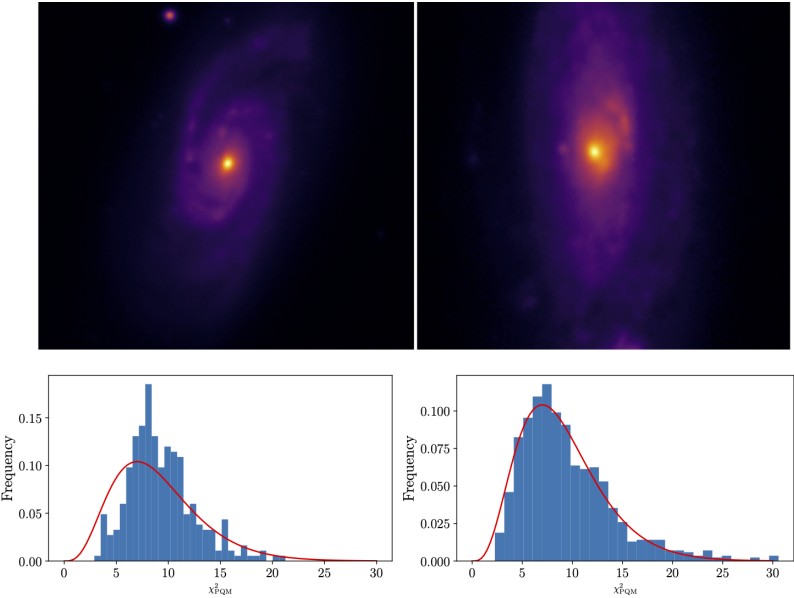

Figure 17: On the top left, a real galaxy image from the probes dataset. On the top right, an image generated using a score-based model. The bottom left plots show the distribution of $\chi^2_{\text{PQM}}$ for 10 and 100 regions, which as expected follow the correct chi-squared distribution. The botton right plots show the same distributions for the generated images. We see that the chi-squared is close to the expected one, meaning the generated images are nearly indistinguishable from real ones. For these experiments, $\chi^2_{\text{PQM}}$ was evaluated by retessellating and resampling the data sets.

between entries and compute distances between the encodings. We show the value of $\chi^2_{\text{PQM}}$ in Fig. 16. We see that $\chi^2_{\text{PQM}}$ goes down as training progresses, until eventually plateauing, as expected.

### F.5 ASTROPHYSICS EXPERIMENTS

As an example of PQMass applied to a scientific problem, we test the method on a problem from astrophysics. We trained a score-based model (SBM) on 256x256 real galaxy images from the Probes dataset, which is a compendium of 3163 high-quality local late-type galaxies Stone & Courteau (2019). Since there has been increasing interest in the use of SBMs as high-dimensional priors for inference problems, the question of the accuracy (in the statistical sense) of the trained SBM is very important. Using the PQMass metric, we were able to establish the probability that the generated samples come from the same underlying distribution as the training samples. On 256x256 images,

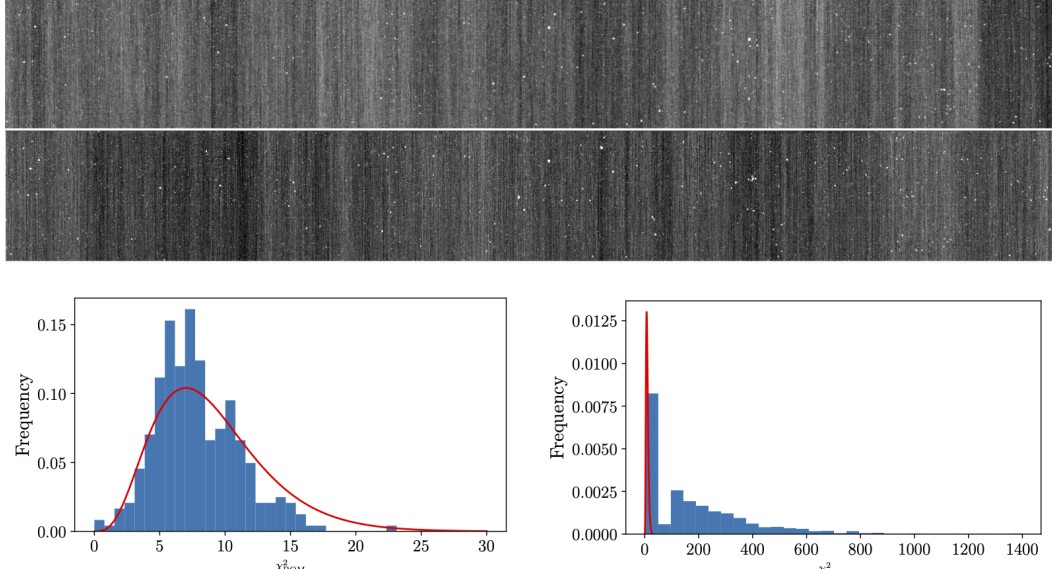

Figure 18: Example of a true dark image (top) and generated dark (middle). The bottom left plot shows the distribution of $\chi^2_{\text{PQM}}$ for 10 regions when comparing half of the 6960 ground truth darks with the other half, while the bottom right compares 2000 generated dark samples from the SBM with the full set of the real darks. The two sets are clearly out of distribution, as the generative model has not learned the true distribution of hot pixels, despite the evident similarity by eye between the ground truth and generated samples. For these experiments, $\chi^2_{\text{PQM}}$ was evaluated by retessellating and resampling the data sets.

PQMass runs in less than one hour for 1000 evaluations on a single Nvidia A100 GPU (about 3s for every $\chi^2$ evaluation). When we compare a random split of the probes dataset with itself, the $\chi^2_{\text{PQM}}$ values are distributed according to the expected $\chi^2$ distribution while we can reproduce a curve similar to the one presented in Fig. 5 as a function of training iteration. The results are shown in Fig. 17.

Next, another scientific application of interest is using SBMs to model noise likelihoods as described in Legin et al. (2023). In this case, the accuracy of the learned generative model is of paramount importance. Here, we used the case of pixel darks (images of pure noise) from the Near Infrared Imager and Slitless Spectrograph (NIRISS) aboard the James Webb Space Telescope (JWST) as the noise to be modeled. This instrument has Gaussian noise as well as multiple sources of non-Gaussian additive noise, including dead pixels, faint and saturating cosmic rays, bleeding, a zero bias, blotches, and $1/f$ noise. The lattermost source of noise is especially difficult to completely correct for using traditional methods (Albert et al., 2023), effectively reducing the usable information of scientific images, especially for low signal-to-noise ratios. As an additional experiment, we have trained a SBM on 4000 of the full set of 6960 real 2048x256 pixel darks. We show examples of the ground truth images and generated images in Fig. 18 as well as two PQMass comparisons, one between two sets of the ground truth and another between the generated images and the ground truth. While the $\chi^2_{\text{PQM}}$ metric can clearly show that half of the real dark samples are in distribution with the other half (bottom left plot in Fig. 18) using 10 regions, it can also detect that the generated samples are not in distribution with real dark samples, despite their visually evident resemblance. Upon further investigation, this can be confirmed by the realization that the correct placement of hot pixels has not yet been fully learned by the generative model.

We now compare PQMass against other standard metrics in the fields, Feature Likelihood Divergence (FLD, Jiralerspong et al. (2023)) and Fréchet Inception Distance (FID, Heusel et al. (2017)). We utilize two astrophysics datasets introduced earlier—Probes and JWST—and introduce a third dataset, SKIRT TNG (Bottrell et al. (2024)), which consists of simulated galaxy data. We use the SBM trained by Missael Barco et al. (2024). We compare samples generated from the respective SBM against the validation set, the ground truth. Additionally, we conduct a null test where, for each dataset, we split

Table 5: Comparison of PQMass, FLD, and FID metrics across various astrophysical datasets. PQMass is configured with 10 regions, yielding an expected score of approximately 9 for in-distribution comparisons. For PQMass, scores significantly above 9 indicate out-of-distribution samples. All PQMass experiments were done by retessellating and resampling the sets for evaluating $\chi^2_{\text{PQM}}$. For FLD and FID, lower scores generally suggest more similar distributions. "Samples vs Ground Truth" compares generated samples to the original dataset, while "Ground Truth vs Ground Truth" compares two subsets of the original data to assess metric consistency.

| Dataset | Dimensionality | Num of Samples | PQMass | FLD | FID |
|---|---|---|---|---|---|
| Probes (Samples vs Ground Truth) | 3x256x256 | 1000 vs 2059 | 10.97 | 52.13 | 594.94 |
| Probes (Ground Truth vs Ground Truth) | 3x256x256 | 1029 vs 1030 | 9.42 | −21.72 | 84.64 |
| SKIRT TNG (Samples vs Ground Truth) | 3x64x64 | 1500 vs 2549 | 9.84 | 24.29 | 613.74 |
| SKIRT TNG (Ground Truth vs Ground Truth) | 3x64x64 | 1230 vs 1229 | 9.83 | −4.09 | 12.14 |
| JWST (Samples vs Ground Truth) | 1x256x2048 | 3480 vs 3480 | 131.70 | 11.08 | 128.59 |
| JWST (Ground Truth vs Ground Truth) | 1x256x2048 | 2000 vs 6969 | 7.88 | −3.06 | 1.07 |

the validation set into two subsets to test whether the metrics can detect that the subsets come from the same distribution. We highlight the dimensionality of each dataset, with JWST being the highest dimensional experiment.

We showcase our results in Table 5, where we use 10 regions for all experiments. In the null test, PQMass is consistently able to detect that they come from the same underlying distribution; however, FLD struggles, returning negative values indicating a measure of duplicates in the two datasets, which we know to not be true. FID struggles with Probes and SKIRT TNG but can showcase that for the JWST example, it is in distribution. Furthermore, when comparing samples generated from the SBM to the validation test, PQMass can detect that the samples come from the same distribution for Probes and SKIRT TNG, which we know to be correct. It can also capture the out-of-distribution nature of the JWST samples, which is correct. However, the same cannot be said for FLD or FID. For Probes and SKIRT TNG, FLD and FID claim the samples are out of distribution, and for JWST, FLD claims that the samples are in distribution, whereas FID correctly claims that the samples are out of distribution. Here, we showcase that PQMass can detect if the two datasets are in or out of distribution consistently where other standard metrics in the field either fail or are highly inconsistent.

### F.6 EXPERIMENT ON PROTEIN SEQUENCES

As another experiment on a real dataset with a different structure, we showcase the ability of PQMass to detect differences in datasets of protein sequences. As our example, we use the ESM Metagenomic Atlas dataset (Lin et al., 2023)[7]. We perform two comparisons: In the first one, we randomly split the dataset into two, and repeat the null test shown in §3.1 and §A. As our distance metric, we use the Levenshtein, or edit distance (*i.e.*, the number of edits needed to turn one sequence into the other). The result is shown in the blue histogram of Fig. 19. This shows not only that our method can also work in this data format, but also that it can work with a different distance metric.

Secondly, we split the dataset into two: One with Protein post-translational modification (PTM) higher than 0.5, and one with PTM lower than 0.5. This is equivalent to splitting the dataset into one of high confidence protein sequences, and one of low confidence. We then repeat the test and show the result in the orange histogram of Fig. 19. We can clearly see that PQMass can detect that these two datasets come from different distributions. It is a non-trivial result that, by comparing edit distances in sequence space, we can detect that the high PTM and low PTM sequences belong to different distributions.

### G PQMASS IN DIFFERENT METRICS

The PQMass test is performed using only distances between sample points, thus one may choose any distance metric, or kernel (representation) function, to perform the test. The choice of metric may affect what features PQMass is most sensitive to. In Fig. 20 we show how PQMass performs on a toy problem using various distance metrics. A Gaussian mixture model with 20 components is created in 100 dimensions, for each component the means are drawn from a uniform distribution with range $(-10, 10)$ in all dimensions, the covariance matrices are constructed with eigenvectors aligned randomly and eigenvalues drawn from a log-uniform distribution with range $(-1, 1)$. The

---

[7] https://esmatlas.com

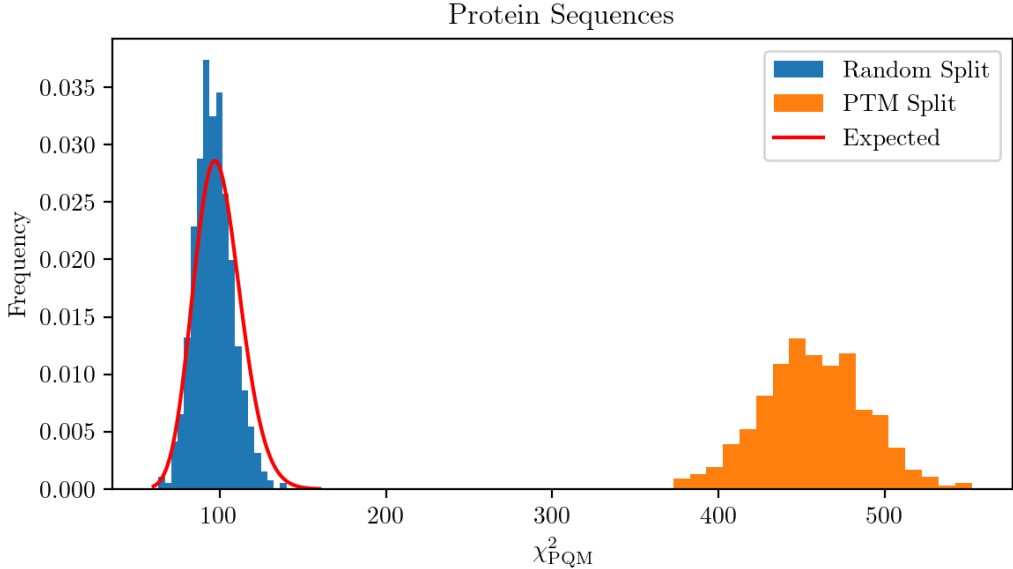

Figure 19: Experiment with protein sequences. In blue, we compare two randomly chosen subsets of the ESM metagenomic Atlas dataset. In orange, we compare high PTM to low PTM sequences. PQMass can clearly detect that the random splits are samples from the same distribution, while the high and low PQM proteins correspond to two different datasets. This experiment was done by retessellating and resampling the sets for evaluating $\chi^2_{\text{PQM}}$.

weight for each component is also drawn from a log-uniform distribution with range $(-1, 1)$. For each test, we draw two sets of $2^{12}$ samples from the GMM as input to PQMass. The null test takes these inputs as-is, for the "scale" out-of-distribution test the samples are scaled by 1.08, and for the "rotated" out-of-distribution test the samples are rotated by approximately 0.08 radians on a random axis (scale and rotation chosen arbitrarily for figure quality). To generate the histograms in the figure, the tests are re-run $2^{14}$ times each.

All metrics in Fig. 20 correctly fail to reject the null test (left subplot), as should be expected.

The only "failure mode" for PQMass is to not reject the null when the two distributions are different because *the metric is degenerate* (*i.e.*, is really a pseudometric). This occurs in the centre subplot, where *correlation* and *cosine* metrics fail to reject the null because they are not sensitive to rescaling. In the right subplot, where we use a small rotation as the perturbation, the *correlation* and *cosine* are, in fact, the most sensitive to this change, demonstrating how a choice of metric impacts the sensitivity to differences in the two samples.

We also repeat the experiment described in §3.3, where we evaluate the impact of various distance metrics on PQMass as Gaussian noise with increasing variance is added to the ImageNet dataset. In Figure Fig. 21, we show that when no noise is added (null test), all distance metrics will correctly detect that the images are in distribution (left subplot). We also show that as we add noise, different metrics showcase different sensitivity and thus affect $\chi^2_{\text{PQM}}$ differently. In both tests (and indeed all our tests) the Euclidean distance – the one used in the main experiments – is among the most discriminating.

## H    COMPARISON OF PQMASS TO OTHER TWO-SAMPLE TESTS

Here, we compare PQMass to other two-sample tests. One such class of tests is described in Schilling (1986), which proposes a nonparametric approach to determine whether two multivariate samples of points are drawn from the same underlying distribution. In the proposed class of tests, the statistic used is calculated as the fraction of nearest-neighbor comparisons where both the point and its nearest neighbor are from the same sample. Here, we consider two specific variations: an unweighted version,

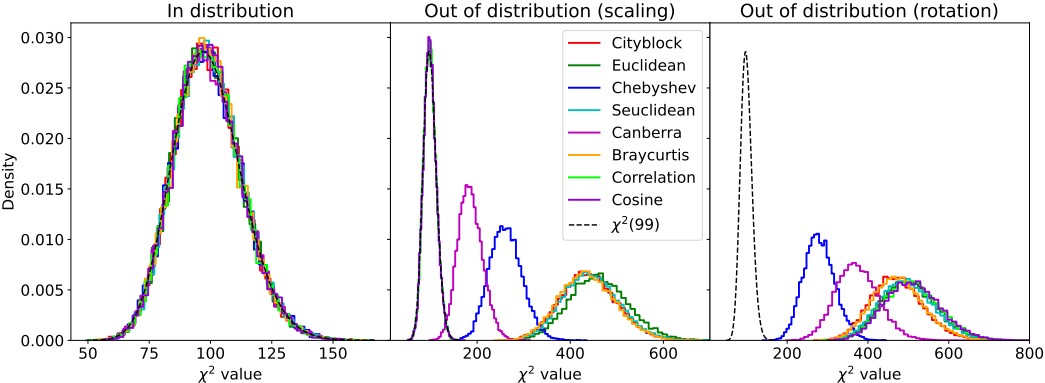

Figure 20: Comparison of PQMass applied with various metrics. The metrics are described in detail in the `scipy.cdist` documentation; the first three correspond to standard $L_1$, $L_2$, and $L_{inf}$. $\chi^2$ distributions are presented in each figure. Left: null test to show any metric will return a null result in the null case. Middle: out-of-distribution example where samples are scaled by a small amount. Right: out-of-distribution example where samples are rotated by a small amount.

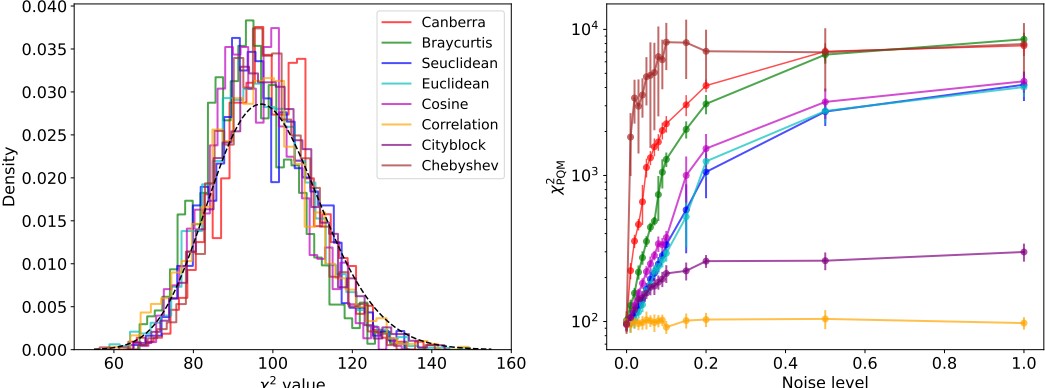

Figure 21: Comparison of the performance of various distance metrics to detect the addition of Gaussian noise with increasing variance noise to ImageNet data. *Left*: Null test with no noise added, where all metrics correctly identify the data as in-distribution. *Right*: Gaussian noise with progressively larger variance added to one of the sets affects $\chi^2_{\text{PQM}}$ differently depending on the metric used. For this experiment, $\chi^2_{\text{PQM}}$ was evaluated by retessellating and resampling the datasets.

Table 6: Two multivariate two-sample tests from Schilling (1986) (unweighted and weighted) applied to two sets of 50 samples from $\mathcal{N}(\mathbf{0}, \mathbf{I})$ and $\mathcal{N}(\mathbf{0.5}, \mathbf{I})$. We wish to assess whether these tests can detect that the two samples are out of distribution. For each test, we display the expected in-distribution value and the obtained result, as well as the corresponding p-value. In both cases, the results are statistically very close to the expected values, indicating no detection that the two distributions are out of distribution.

| Metric | Expected Result | Actual Result | p-value |
|---|---|---|---|
| $T_{k,n}$ | 49.4900 | 52.4000 | 0.5596 |
| $U_{k,n,w}$ | 1.1300 | 1.1245 | 0.7055 |

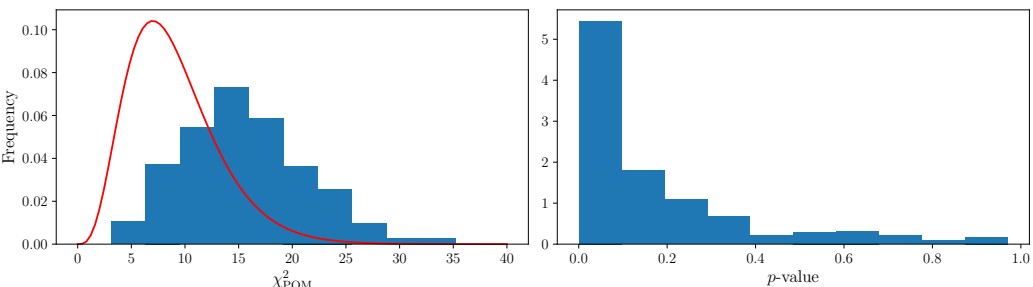

Figure 22: Comparison of 50 samples from $\mathcal{N}(\mathbf{0}, \mathbf{I})$ to 50 samples from $\mathcal{N}(\mathbf{0.5}, \mathbf{I})$, with PQMass. Unlike the multivariate two-sample test shown in Table 6, PQMass can confidently detect that the two sets of samples are indeed not from the same distribution, showcasing that PQMass is more sensitive than the multivariate two-sample tests presented in the main text.

denoted $T_{k,n}$, which evaluates the proportion of nearest neighbors that are similar, and a weighted version, denoted $U_{k,n,w}$, where distances are assigned decreasing weights with increasing distance.

We show that PQMass can outperform this methodology by comparing the performance of the unweighted and weighted nearest-neighbor two-sample tests against PQMass. Specifically, we consider two distributions, $\mathcal{N}(\mathbf{0}, \mathbf{I})$ and $\mathcal{N}(\mathbf{0.5}, \mathbf{I})$, where $\mathbf{0} = (0, 0)$ and $\mathbf{0.5} = (0.5, 0.5)$ are 2-dimensional mean vectors and $\mathbf{I}$ denotes the $2x2$ identity matrix. Samples from these distributions should be identified as out-of-distribution relative to each other. Using 50 samples from each distribution, we evaluate $T_{k,n}$ and $U_{k,n,w}$ in Table 6. For the unweighted test statistics, the expected value is $T_{k,n} = 49.4900$, and the observed value is $T_{k,n} = 52.4000$, with a p-value of p-value$_{T_{k,n}} = 0.5596$. These results indicate that the unweighted two-sample test fails to reject the null hypothesis and incorrectly suggests the distributions are in-distribution. For the weighted two-sample test, the expected $U_{k,n,w} = 1.1300$, the observed value is $U_{k,n,w} = 1.1245$, and the p-value is p-value$_{U_{k,n,w}} = 0.7055$. Again, the results do not indicate a significant difference between the distributions. In contrast, as shown in Fig. 22, when running with PQMass using 10 regions (due to limited samples), the resulting $\chi^2_{\text{PQM}}$ as well as the p-value$_{\text{PQM}}$, confidently indicate the data does not come from the same underlying distribution.

## I  PERMUTATION TESTS

When PQMass is run for a single iteration (single tessellation) it is an exact statistical test, however the power of this test is considerably dependent on the particular arrangement of the Voronoi bins. To increase the power of the test, we repeat the test multiple times and verify that $\chi^2_{PQM}$ is $\chi^2$ distributed. In cases where data samples are limited, however, running multiple tests by ressampling and retessellating can be difficult. In those cases, in particular in some of our experiments, we can run many retessellations on the same sets of samples to maximize the use of existing samples and ensure we encounter especially discriminating configurations.

In this case, however, since the samples in the retessellations are the same, there is some shared information between tests and the $\chi^2_{PQM}$ values from these retessellations must no longer follow a $\chi^2$ distribution under the null hypothesis. This presents a challenge for understanding the meaning of a

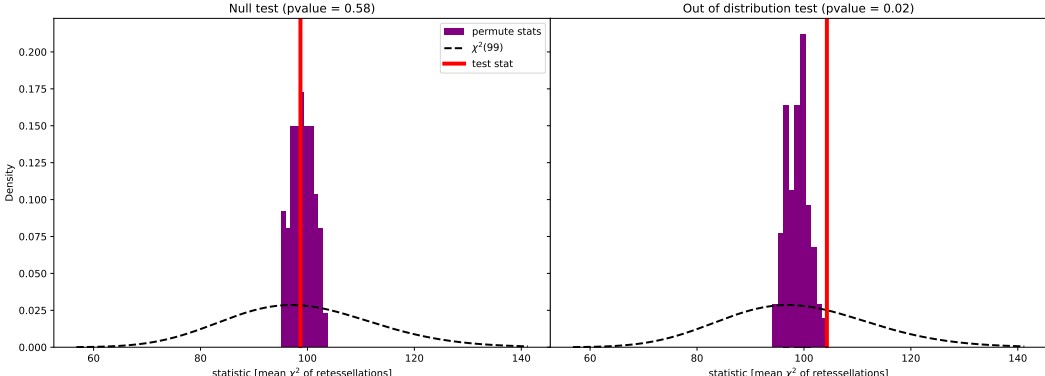

Figure 23: Permutation test for validity of taking the mean of many PQMass retessellations. Samples are drawn from a 100 dimension standard Gaussian, we use 1000 samples for each input. Shown are the $\chi^2(99)$ distribution (black dashed), mean of many retessellations (red vertical line), and the permutation test (purple histogram). **Left**: A null test which, as expected, fails to reject the null hypothesis. **Right**: An out of distribution test where one sample has been scaled by 1.1. The null hypothesis is indeed rejected.

PQMass $\chi^2$ retessellation histogram. This issue can be easily eliminated via a permutation test (Good, 2013).

Any measurement on some dataset can be turned into an exact test on the null hypothesis that two samples are drawn from the same distribution using a permutation test. In brief, a permutation test involves taking some scalar measurement on two datasets, lets say PQM($x, y$) where $x$ and $y$ are the two samples, and we re-run the measurement on a permutation of the datasets PQM($x', y'$) where $x'$ and $y'$ are a random permutation of the combined dataset $\{x, y\}$. The permuted samples are thus drawn (without replacement) from a mixture distribution of the two samples. Running many permutations one builds up a distribution of values for PQM($x', y'$) against which we may compare the original value PQM($x, y$) to find a p-value. At an intuitive level, this test determines how unique the original arrangement of $x, y$ samples are under the measurement PQM.

The PQMass test is especially sensitive to the differences in two samples, making it an ideal choice for such a test. Specifically, we may now take PQMass run with many retessellations, take the mean of the reported $\chi^2$ values, and use this mean as our measurement on the two samples; now fully utilizing the discriminating power of the retessellations in an exact test. We perform a permutation test using PQMass in Fig. 23. The histogram shows PQMass run on random permutations of the input samples ("permute stats" in the figure) while the vertical line shows the result for the input samples ("test stat" in the figure). In the null case, we achieve a p-value consistent with $U(0, 1)$ while in the out-of-distribution case we reject the null with a low p-value.

In both cases, the mean of many re-retessellations on the original sample would have placed us well within the $\chi^2(99)$ distribution, showing how the permutation test is more powerful at rejecting the null, if such extra power is needed.

In practice, we find that, for high-dimensional problems, the retessellations without resampling very closely approximate a $\chi^2$ distribution (in the null case), though this is not the case in low dimensions. Since it is common in high-dimensional spaces for PQMass to be highly discriminating, it is often not necessary to perform the permutation test to reject the null; one can conservatively do so using the full $\chi^2$ distribution.

If, after a few retessellations, one finds that they cannot easily reject the null, then performing a permutation test can add more discrimination power at the cost of extra compute. Even with this extra computational cost, PQMass remains computationally feasible on large and high-dimensional datasets. We note again that with a permutation test it is possible to turn any metric on two samples into an exact two-sample test; for example, the energy distance may be used (Székely et al., 2004), but the computational complexity of that test is higher than PQMass.

