# OpenReview forum: "PQMass: Probabilistic Assessment of the Quality of Generative Models using Probability Mass Estimation"
_ICLR.cc/2025/Conference — ICLR 2025 Poster_

### Official Review · Reviewer_b1WN · 2024-10-29

**Soundness:** 3
**Presentation:** 3
**Contribution:** 3
**Rating:** 6
**Confidence:** 2

**Summary:**

The article proposes a new method for comparing samples from two distributions, providing a hypothesis test for whether they are identically distributed. The method introduces a Voronoi tessellation, dividing the data space into non-overlapping regions, and then assumes a multinomial likelihood describing the prevalence of observed data in each of the regions: this likelihood is then used to develop a hypothesis test as to whether each distribution has the same underlying probabilities of generating data in each region, which then implies equivalence of the distributions with some assumptions. Some theoretical results are derived to strengthen this argument. The method is then tested on several empirical problems.

**Strengths:**

The idea is pleasingly simple for a relevant contemporary challenge in stats/ML, and explained pretty clearly. The experimental results are strong and quite convincing, especially in higher-dimensional problems without an obvious feature representation.
The figure quality is high throughout, which is a good thing in general.
The experiments themselves are pretty thorough, as are the descriptions of the experiments, especially the astrophysics experiments that might be hard for a non-expert audience. The example detecting the small additive noise is nice in Appendix F.2. Demonstrating the method can find an effect invisible to the naked eye is good.

**Weaknesses:**

The very high figure quality has the probably unintended consequence that the resulting pdf file size is very large (>17MB), which makes it kind of a pain to handle by conventional means. If there were some way to reduce this without overly compromising readability, then that would be good. I suspect that you have one or two figures in the appendix that are unnecessarily high resolution when printed out, but I haven’t figured out which ones (possibly figures 17/18).

A few small language issues:

“Miscrepancy” line 356. I assume you meant “discrepancy”? English in general pretty good.
“The instrument has multiple sources of non-Gaussian additive noise, including… Gaussian noise” is kind of odd.

Figure design:

Figure 6 is hard to understand immediately. The claim about “strongly correlate” (line 446) does not jump out intuitively from the data plotted.

Some of the descriptions of the method’s properties are a little overly flattering:

You call this a “Likelihood-free method”... but, you do use a multinomial distribution.
Desiderata 1: “PQMass works with the intervals of pdfs, it does not make assumptions about the true underlying density field”. The cdf is defined by the pdf, and assumptions about one will imply assumptions about the other, so I don’t think this argument is particularly strong.

“These results show that, at least for “well-behaved” densities, no information is lost by the PQMass statistics”
Did you not say at the start that you didn’t make assumptions about the underlying density?
“No information is lost” in what sense? Clearly information is lost when you quantise. Is this claim meant in some asymptotic limit?

More explanation of the 2-d Gaussian mixture model and the 10-d funnel distribution would be good rather than just a reference, if you can find the space. Appendix is always an option.

**Questions:**

Methodological questions:

Is there any particular reason that you pursued Pearson’s chi squared test ahead of Fisher’s exact test, which should be less vulnerable to small contingency cell counts, which could easily happen here with the randomised voronoi tessellation?

Choice of centres:

Is it really hard to do better than random? Some sort of relatively simple diversity measure would potentially get you quite far here.
“Sample the reference points from a uniform mixture of the empirical distributions of samples from p and q”

What if there are more samples available from one distribution?

What if one of the distributions is known to be more accurate representation of reality and therefore give a better coverage of realistic regions of probability space?

“Perform the test multiple times”? How many is necessary? What are the computation implications?

“This effectively marginalizes the choice of tessellation”: this comment makes me wonder whether a Bayesian formulation of the problem would work well, since you are effectively having sample the latent allocation of the points to reference centres vs cell populations. There will be analytical conjugate Bayes solutions to the multinomial distribution comparison too, so there should be no more pesky posterior sampling in addition to what is being done already.

Data balance between distributions:

What if there are unbalanced numbers of samples available from each distribution? This will be quite common, since in general distributions are not equally easy to sample from. This is especially true if comparing simulated data and real data, or complex vs simpler models. How would this influence the sampling of reference points?

Limitations: good for you for being open “if the number of samples is too small”. Do you have any idea how small? Does it scale with p in a demonstrable way?

---

> ### Author Response · Authors · 2024-11-22
> **Response (1/3)**
>
> Thank you for your review and very detailed comments. We have attempted to answer each of them below.
>
> > The idea is pleasingly simple for a relevant contemporary challenge in stats/ML, and explained pretty clearly. The experimental results are strong and quite convincing, especially in higher-dimensional problems without an obvious feature representation. The figure quality is high throughout, which is a good thing in general. The experiments themselves are pretty thorough, as are the descriptions of the experiments, especially the astrophysics experiments that might be hard for a non-expert audience. The example detecting the small additive noise is nice in Appendix F.2. Demonstrating the method can find an effect invisible to the naked eye is good.
>
> We appreciate the reviewers kind words, and are very grateful for their feedback
>
> > The very high figure quality has the probably unintended consequence that the resulting pdf file size is very large (>17MB), which makes it kind of a pain to handle by conventional means. If there were some way to reduce this without overly compromising readability, then that would be good. I suspect that you have one or two figures in the appendix that are unnecessarily high resolution when printed out, but I haven’t figured out which ones (possibly figures 17/18)
>
> We apologize for our oversight. It appears that figures 17 and 18 were indeed partly to blame, as well as 12. We have reduced the size in the updated version.
>
> > A few small language issues: “Miscrepancy” line 356. I assume you meant “discrepancy”? English in general pretty good. “The instrument has multiple sources of non-Gaussian additive noise, including… Gaussian noise” is kind of odd.
>
> These are, indeed, grammatical errors. We thank the reviewer for pointing them out, and have corrected them.
>
>
> > Figure design Figure 6 is hard to understand immediately. The claim about “strongly correlate” (line 446) does not jump out intuitively from the data plotted.
>
> That is reasonable, this is indeed a busy figure. We have softened the statement about the correlation, since we agree that while the correlation is visible, the word "strong" might be exaggerated.
>
> > You call this a “Likelihood-free method”... but, you do use a multinomial distribution
>
> Thank you for the opportunity to clarify this point:
>
> In the context of generative models, "likelihood-free" has been used to mean simply "not using the density of or estimated by the generative model" (but instead using statistics computed from samples, as PQMass does). Likelihood-free tests, such as PQMass, FID, etc., can be used with generative models where exact density evaluation is intractable, such as GANs and diffusion models, and are thus favored over data log-likelihood in many applications.
>
> This is distinct from another meaning of "likelihood-free" in machine learning, which is related to simulation-based inference (i.e., posterior estimation where joint probabilities are not available but ancestral sampling is possible).
>
> In neither case does the term imply that probability computations are not involved in the test.
>
> > “PQMass works with the intervals of pdfs, it does not make assumptions about the true underlying density field”. I mean, the cdf is defined by the pdf, and assumptions about one will imply assumptions about the other, so I don’t think this argument is particularly strong
>
> We suspect this may be a propagation of the small misunderstanding from the previous point. We simply meant that PQMass does not make assumptions about the approximability of the two test distributions by some functional form, as some other methods do (e.g., FLD relies on a Gaussian mixture approximation and FID on a Gaussian approximation).
>
> > “These results show that, at least for “well-behaved” densities, no information is lost by the PQMass statistics” Did you not say at the start that you didn’t make assumptions about the underlying density?
>
> The lack of assumptions refers to approximability (e.g., by Gaussians or Gaussian mixtures). The assumptions of smooth density and full-support reference point distribution are very weak (and can probably be relaxed further). For example, they hold for the convolution of any distribution with a small Gaussian. We will revise the statement to make clear what is meant.
>
> > “No information is lost” in what sense? Clearly information is lost when you quantise. Is this claim meant in some asymptotic limit?
>
> This means that if two distributions are distinct, then the test has a positive probability of distinguishing them (and the probability approaches 1 in the asymptotic limit).
>
> This contrasts with tests that approximate the distributions by a certain fixed class. For example, if two distributions have the same second-moment statistics in feature space, FID will **not** distinguish them, even in the asymptotic limit of infinite samples, while PQMass will.

---

> ### Author Response · Authors · 2024-11-22
> **Response (2/3)**
>
> > More explanation of the 2-d Gaussian mixture model and the 10-d funnel distribution would be good rather than just a reference, if you can find the space. Appendix is always an option.
>
> These are both synthetic distributions used for benchmarking sampling methods. They are chosen because of the multimodality of the first, and the difficult to model shape of the second. We have clarified this in Appendix C, which also shows plots of both distributions.
>
> > Methodological questions: Is there any particular reason that you pursued Pearson’s chi squared test ahead of Fisher’s exact test, which should be less vulnerable to small contingency cell counts, which could easily happen here with the randomised voronoi tessellation?
>
> The main reason is that Fisher's exact test, at least traditionally, is used on $2\times2$ contingency tables, whereas we have $N\times2$ tables. As far as we are aware, extending Fisher's test to an $N\times2$ case would require computing a combinatorially large number of $2\times2$ tests.
>
> > Choice of centres Is it really hard to do better than random? Some sort of relatively simple diversity measure would get you quite far here, i reckon “Sample the reference points from a uniform mixture of the empirical distributions of samples from p and q”
>
> We have tested a variety of sampling schemes for the reference points. Our ultimate choice of sampling reference points from the mixture of p and q is reliably efficient and avoids pathological cases (see the second table included in the response to Reviewer 6iXh).
>
> > What if one of the distributions is known to be more accurate representation of reality and therefore give a better coverage of realistic regions of probability space?
>
> Our method simply estimates the probability that both sets of samples are drawn from the same distribution, and therefore has no perception of reality, or ground truth. If one of the sets of samples has better coverage than the other, and this difference is significant enough, our method can point out that these samples come from different distributions.
>
> > “Perform the test multiple times”? How many is necessary? What are the computation implications? “
>
> More repetitions give a better sense of the uncertainty in the estimation, but there is an obvious linear scaling of computational costs with the number of repetitions. In our experiments, we empirically found 20 to be a good enough number to estimate this uncertainty.
>
> > “This effectively marginalizes the choice of tessellation”: this comment makes me wonder whether a Bayesian formulation of the problem would work well, since you are effectively having sample the latent allocation of the points to reference centres vs cell populations. There will be analytical conjugate Bayes solutions to the multinomial distribution comparison too, so there should be no more pesky posterior sampling in addition to what is being done already
>
> Indeed, if we assume a ${\rm Dirichlet}(\alpha)$ prior over the probability masses of $n_R$ regions in a tessellation, and observe count vector $k_P=(k_{P,i})_{i=1}^{n_R}$ of points in these regions, the posterior distribution over the masses is Dirichlet-multinomial.
>
> The posterior predictive probability of observing an independent sample of points with count vector $k_Q=(k_{Q,i})_{i=1}^{n_R}$ is then
>
> ${P}({k}_Q\mid {k}_P)$
>
> $=$
>
> $\frac{\Gamma(n_R\alpha+m)\Gamma(n+1)}{\Gamma(n+n_R\alpha+m)}\prod_{i=1}^{n_R}\frac{\Gamma(\alpha+k_{P,i}+k_{Q,i})}{\Gamma(\alpha+k_{P,i})\Gamma(k_{Q,i}+1)},$
>
> where $m=\sum_ik_{P,i}$, $n=\sum_ik_{Q,i}$ are the numbers of samples.
>
> Although this gives us posterior probabilities, it leaves us with the problem of choosing $\alpha$, as the properties of the distribution from which the reference points defining the regions are drawn will affect the distribution of masses.

---

> > ### Author Response · Authors · 2024-11-22
> > **Response (3/3)**
> >
> > > Data balance between distributions What if there are unbalanced numbers of samples available from each distribution? This will be quite common, since in general distributions are not equally easy to sample from. Especially true if comparing simulated data and real data, or complex vs simpler models. How would this influence the sampling of reference points?
> >
> > Note that our method makes no assumptions about the number of samples per distribution, or their ratio, and in fact some of the experiments (e.g. Appendix F4) have different numbers of samples in each distribution.
> >
> > > How would this influence the sampling of reference points?
> >
> > PQMass selects reference points proportionally to the number of samples in each dataset, ensuring that smaller datasets contribute fewer reference points thus preserving as much information in the smaller dataset as possible.
> >
> > > Limitations: good for you for being open “if the number of samples is too small”. Any idea how small? Does it scale with p in a demonstrable way?
> >
> > It will vary depending on the data and it's complexity (e.g. how many modes it has in the case of a multimodal distribution, as well as their relative mass) so it is not easy to determine "how many samples is enough." We have noticed that for 100 reference points, at least 1000 samples is a good lower limit but this is not always the case. In many applications, we have seen that using $n_R\approx\sqrt{N}$ where $N$ is the number of samples in the smaller set yields the best results, but this would require further exploration to be formalized. This is shown, for example, in the new version of Figure 2 included in the paper, where 100 reference points are used and at least 1000 samples per set are needed to see an almost perfect agreement between the $\chi^2_{PQM}$ and the ideal distribution.
> >
> >
> > > What is the relevance of this work to likelihood-free inference? Working with samples from simulators and comparing with observed data, then needing to perform inference over simulator parameters.
> >
> > While PQMass is not an inference method, we can see multiple use cases for such a method in **validating** a simulation-based inference pipeline. Typically, in SBI frameworks, a simulator is used to produce mock observations of observed data. A first use-case example would be where different repeated real experiments are expected to sample the parameters of the simulator (which we want to infer) according to the prior. Then, the distribution of the simulations and the distribution of the real data set should be coming from the same data-generating distribution **if** the physical model and the prior are not misspecified. PQMass can reveal potential model misspecifications by comparing these distributions—a task that becomes especially challenging in high-dimensional data, where summary statistics are often insufficient.
> >
> > There are multiple other uses of PQMass as a validation method in SBI. In such frameworks, while the use of a generative model is not essential, there are many scenarios where an emulator is used to produce fast simulations in sufficient quantity. It is typical to have an emulator for expensive components of the simulation pipeline (e.g. hydrodynamical simulations in cosmology, or score-based models for simulating detector noise).  PQMass can evaluate emulator accuracy by verifying whether their outputs align with the distributions produced by the original, more expensive simulator.

---

> > > ### Comment · Reviewer_b1WN · 2024-11-25
> > >
> > > Thank you for the clarifications offered. I will take them into consideration when determining my final score.

---

> > > > ### Author Response · Authors · 2024-12-03
> > > >
> > > > We would like to thank the reviewer for their time and efforts in engaging with our paper during the refereeing process, and we believe the quality and clarity of our paper have been improved as a result. As the deadline is approaching, we would like to remind the reviewer to update their final decision on the paper's score as they deem appropriate.

---

### Official Review · Reviewer_mbPi · 2024-11-01

**Soundness:** 2
**Presentation:** 3
**Contribution:** 2
**Rating:** 6
**Confidence:** 4

**Summary:**

The authors proposed a simple method for comparing the distribution $q$ of a generative model with the true data distribution $p$. The proposed method first discretizes the data space and then compare the proportion of data points in each region. Since this becomes a distribution comparison for a multinomial distribution, it can be tested using Pearson’s $\chi^2$ test. If the test indicates a difference between the two distributions, one can conclude that $p \neq q$, making this method a sufficient condition for identifying distribution differences. The authors demonstrated the effectiveness of the proposed method on synthetic data as well as datasets such as MNIST and CIFAR-10.

**Strengths:**

The strength of the proposed method is in its simplicity. After discretizing the data space, one only needs to count the data points in each region and calculate the test statistic. This makes the method computationally much more efficient compared to existing distribution comparison methods, such as MMD.

**Weaknesses:**

A weakness of this study is the limited comparison with other two-sample test methods. The authors simplified the problem by dividing the data space and reducing it to a multinomial distribution comparison. However, various other methods have been proposed for distribution comparison not only MMD and W2. For example, [Ref1] explores distribution comparison using classifiers and its application to GAN evaluation, while simpler approaches, like those using nearest neighbors, are also available [Ref2]. Although the simplicity of the proposed method is an advantage, comparing its performance with these existing distribution comparison methods would be essential for evaluating its practical utility.

* [Ref1] Revisiting Classifier Two-Sample Tests, ICLR 2017.
* [Ref2] Multivariate Two-Sample Tests Based on Nearest Neighbors, Journal of the American Statistical Association, 1986.

**Questions:**

What are the strengths or advantages of the proposed method compared to other distribution comparison techniques beyond MMD and W2?

---

> ### Author Response · Authors · 2024-11-22
> **Response**
>
> We thank the reviewer for their helpful comments.
>
> > A weakness of this study is the limited comparison with other two-sample test methods. The authors simplified the problem by dividing the data space and reducing it to a multinomial distribution comparison. However, various other methods have been proposed for distribution comparison not only MMD and W2. For example, [Ref1] explores distribution comparison using classifiers and its application to GAN evaluation, while simpler approaches, like those using nearest neighbors, are also available [Ref2]. Although the simplicity of the proposed method is an advantage, comparing its performance with these existing distribution comparison methods would be essential for evaluating its practical utility.
>
> We thank the reviewer for raising this point. We agree that comparison with the nearest neighbors based method is useful, and we have added this to one of our experiments.
>
> We provide an experiment comparing the performance of the nearest-neighbor two-sample tests, both unweighted and weighted, against PQMass. Specifically, we consider two distributions, $\mathcal{N}(0, 1)$ and $\mathcal{N}(0.5, 1)$, where samples from these distributions should be identified as out-of-distribution relative to each other. Using 50 samples from each distribution in two dimensions, we evaluate the unweighted test statistic $T_{k,n}$. The expected value is $T_{k,n} = 49.4900$, and the observed value is $T_{k,n} = 52.4000$, with a p-value of $p_{T_{k, n}} = 0.5596$. These results indicate that the unweighted two-sample test fails to reject the null hypothesis and incorrectly suggests the distributions are in-distribution. For the weighted two-sample test, the expected test statistic is $U_{k, n, w}$ = 1.1300, the observed value is $U_{k, n, w}$ = 1.1245, and the p-value is $p_{U_{k, n, w}}$ = 0.7055. Again, the results do not indicate a significant difference between the distributions. In contrast, when running with PQMass using 10 regions (due to limited samples), the expected $\chi_{PQM}^2$ is 9; however, the actual $\chi_{PQM}^2$ is 15, confidently indicating the data does not come from the same underlying distribution.
>
>
>
> However, we do not believe that comparing with a classifier-based method is the most sensible approach here. The reason for that is that our method is in most cases, used to check the quality of samples from a deep generative model. Therefore, we believe that using a neural network to validate the quality of samples from another neural network is a risky approach (e.g. both generator and test networks may share a failure mode). Such a network based two sample test will share many of the challenges of GANs, and likely raise many further questions such as if one selected the ideal architecture for the discriminator, if it was trained long enough, hyperparameters tuned correctly, and so on. It seems to us that methods based on statistics, such as the one we propose here, and the one introduced in [Ref2] are more appropriate for this problem setting.
>
>
> > What are the strengths or advantages of the proposed method compared to other distribution comparison techniques beyond MMD and W2?
>
> We attempted to address this in section 3.2. The main advantages of the proposed method are better scalability to higher dimensional spaces, and to higher numbers of samples than W2 and Radial Basis Function MMD (Fig. 4) and higher sensitivity to missing modes than Linear Basis Function and MMD (Fig. 3).

---

> > ### Comment · Reviewer_mbPi · 2024-11-25
> > **Re: Response**
> >
> > Thank you very much for the detailed response.
> > The comparison with the nearest neighbor approach looks promising.
> > I would suggest you to add these results in the paper (possibly in appendix).
> > As my concern is addressed, I will raise my score to 6.

---

> > > ### Author Response · Authors · 2024-11-28
> > >
> > > We thank the reviewer for their positive comments. As suggested, we have updated the draft to include an additional appendix (Appendix H) presenting the comparison with the nearest neighbor two-sample tests.

---

### Official Review · Reviewer_6iXh · 2024-11-03

**Soundness:** 3
**Presentation:** 3
**Contribution:** 3
**Rating:** 6
**Confidence:** 4

**Summary:**

The paper proposes a method for testing for the equality of two distributions from which two samples are obtained, by partitioning the sample space and comparing the multinomial vectors of counts of samples in each of the partitions via a chi-squared-distributed test statistic. The approach works for broad classes of data.

EDIT: Following the review/discussion period and (prospective) changes to the manuscript by the authors, I am increasing my score by 1.

**Strengths:**

The paper is well presented and understandable. A broad range of examples and simulations were presented demonstrating fairly broad applicability. The test statistic is a basic and well known statistical quantity, so the method is fast to implement.

**Weaknesses:**

Not clear if this is a weakness or a strength, but the proposed test is a well known, basic statistical test. The contribution of this paper, therefore, is to note that its easy to apply this test to a broad range of data types in order to detect a difference between the generating distribution of two samples. It's not clear if this is substantial enough a contribution for ICLR, though I do not wish to imply that substantial contributions need to involve complex methods.

Consistency Guarantees:
I'm not quite clear on or convinced about some of the claims being made.
* a) Proposition 1: Does the consistency here require any conditions on m and n as they approach -> infinity? E.g., what about their relative convergence rates (does m->infinity faster than n, for example). E.g. what if m/n -> K, where K=infinity, a finite constant, or zero. Presumably K=infinity or K=0 would break things and the test would no longer work (especially as you're sampling finite n_R samples).
* b) Proposition 1: Some clarification on the claim also needed here, as it's conditional on the n_R samples. E.g. if p = t_{\nu_p} and q = t_{\nu_q} (univariate t distributions with different degrees of freedom), and z_1 = -z_2, then you will never distinguish p and q under any rates of m, n -> infinity.
* c) Proposition 2: This proposition is ok, but really this situation won't happen, as the method explicitly requires that the z_i are taken from the set of (x_i, y_j), and so n_R is bound by m+n which are fixed and finite in this proposition (though the method will fail far before this as there will be no samples left to calculate the test statistic), as noted in the discussion. So how useful is Prop 2 in practice, particularly for small m+n?
* d) Proposition 2: "no information is lost by the PQMass statistic." would seem to be a bit of an overextension in light of the above point. But also because for n_R->\infty, the distribution of the statistic under the null would be a chi-squared distribution with infinite degrees of freedom. "No information is lost" only happens in this limit (as otherwise with a finite n_R, information can clearly always be lost), but how does your test statistic compare to its null distribution in this limit?

Practical implementation comments:
* a) As far as I can tell, you've only considered exactly balanced samples each time: m=n. While in some applications the user has control over both m and n, this will not be the case in many scenarios. So how well does the test perform when the two samples are imbalanced, including extreme cases?
* b) Using reference samples z_i randomly taken from the two distributions (x_i) and (y_i) has a long history, even when comparing samples from two MCMC samplers (or the same MCMC sampler twice, when considering convergence diagnostics; e.g. Sisson & Fan (2007). Statistics and Computing, 17, 357-367). So this approach is fine. However, a disadvantage of this is that when p and q differ only, say, by a small mode with low weight in the tail, then by sampling uniformly over the x_i and y_j, one is reasonably likely to miss samples in this mode, as the sampling will naturally favour repeated samples in areas of higher density, and hence the test will miss the obvious difference between p and q. (The examples in Section 3 seemed to consider equally weighted mixtures only ...). Hence the need to repeat the test multiple times with different z_i. Is there any argument to be made to reconsider this equal sampling of the z_i from (x_i) and (y_j), to ensure there is a greater spread over the space, rather than doubling (and tripling, and ...) down on high density regions?
* c) Figure 2: presumably when "this process" is repeated 500 (or 200?) times, this also means resampling the z_i? (Could be clearer in the caption.) But really, this histogram doesn't really seem to follow the red density. Suggest that the number of reps is increased here considerably if you're wanting to claim these distributions are the same. Also the y-axis label of "Frequency" seems to be incorrect, as the axis ticks aren't measured in counts.
* d) In the results in general, the empirical mean of the PQMass statistic (4) seems to be taken as "the statistic" for determining whether p=q (or p!=q). Is it obvious that this mean statistic has a chi-squared(n_R) distribution under the null? Each of the components of this statistic are chi-squared(n_R) (conditional on z_i) under the null, but is it obvious that they are independent from eachother? (chi-squared(m) + chi-squared(n) = chi-squared(m+n) if the two LHS distributions are independent). Certainly the alternative distributions are not the same for different z_i. So can this "mean PQMass statistic" actually be compared to the chi-squared null distribution in any meaningful way? Or is the use of the mean PQMass statistic merely limited to relative value comparisons? In which case, the paper should be clearer about all of this and tone down the chi-squared relevance.
* e) Section 4: Limitations. The method also appears to require independent samples (x_i), (y_i), otherwise the multinomial model does not hold. How likely is this in practice? In the MCMC comparisons in Table 1? Etc.


Typos:
* p.2 para 2, wrong parentheses surrounding reference list starting with Theis et al.
p.5. Algorithm step 1. "choice" -> "chosen"
p.6. Section 3.1. The number of test replicates here is 200, but in Figure 2 it's listed as 500. One of these must be wrong.
p.6. Table 1. D was previously the distance metric between two sample points (c.f. Section 2.2, Algorithm). Now D has switched to number of dimensions. Maybe separate the notation?

**Questions:**

See Weaknesses.

---

> ### Author Response · Authors · 2024-11-22
> **Response (1/4)**
>
> We appreciate the reviewer’s comments and the opportunity to clarify the contributions of the paper. While it is true that the chi-squared test is a well-known statistical method, the specific way that PQMass has been designed to use a chi-squared test to assess the performance of generative models of different modalities in high dimensions is entirely novel.
>
> Recent publications of methods like FID and their growing use demonstrate that there exists a clear and pressing need in the community for such methodology.
>
> PQMass addresses this very need. While PQMass is computationally simple, its simplicity should not be equated to triviality. PQMass is a novel method that provides a reliable, interpretable, broadly applicable, and inexpensive tool to test machine learning models without the use of other machine learning models.
>
> We believe that PQMass can be a highly impactful method. Its publication will certainly help the community learn about this method and allow it to positively contribute to the field.
>
>
> ### On consistency guarantees
>
> a) In Proposition 1, we require $m$ and $n$ to grow at rates such that Fisher's test for multinomial distributions with $m$ and $n$ samples from the two distributions distinguishes them. This is guaranteed by $m/n$ being bounded both above and below by strictly positive, finite scalars. We will clarify this is the meaning of the limit in the text.
>
> b) $\pi$ depends on $r$, so the probability referred to in Proposition 1 is over the choice of $n_R$ samples from $r$. Of course, for "bad" choices of reference points, we may be unlucky and not distinguish $p$ and $q$ (if the two resulting cells have the same mass under $p$ and $q$). The proposition states that with positive probability, this does not happen.
>
> c) Proposition 2 is indeed about the asymptotic limit. Statistical tests comparing a model to ground truth data assume that the latter is sampled i.i.d. from some underlying distribution, even if we only have access to a finite number of samples. The proposition simply tells us that the sensitivity of the test approaches 1 as the number of samples grows, not the rate. The usefulness of the test itself is shown by the empirical evidence in our experiments.
>
> d) We clarify two points:
> - First, "no information is lost" means that if two distributions are distinct, then the test has a positive probability of distinguishing them (and the probability approaches 1 in the asymptotic limit). This contrasts with tests that approximate the distributions by a certain fixed class: for example, if two distributions have the same second-moment statistics in feature space, FID will **not** distinguish them, even in the asymptotic limit of infinite samples, while PQMass will.
> - Second the $n_R\to\infty$ limit in Proposition 2 does **not** refer to performing the test with "$n_R=\infty$" to compare two chi-squared distributions with infinite degrees of freedom. The proposition states that the probability that the tessellation defined using $n_R$ random reference points sampled from $r$ separates $p$ and $q$ -- *which is a well-defined quantity for any $n_R$* -- approaches 1 as $n_R\to\infty$.

---

> > ### Comment · Reviewer_6iXh · 2024-11-29
> >
> > Thank-you for your response. And my apologies for my late response. In terms of points a)-d), I understand your response. What would be good to see, however, is some modification of the paper (either a proposition statement, or a single plain-language comment after the proposition) to explain or clarify these issues a little. It seems that no changes have been made in the revised version of the pdf for any of these point (and I'm looking at the revised version as the caption for Fig 2 has "2^14" in it), and so its quite easy for a reader to reach the same conclusions that I did. Really, communication is helpful for your paper here.
> >
> > For all of the above, I'm not asking these questions because I don't know the answers already. I'm asking them because I think there should be changes in your paper to communicate what you're actually claiming more clearly and more fairly.

---

> > > ### Author Response · Authors · 2024-12-03
> > >
> > > We thank the reviewer for their comments. We apologize for not highlighting the edits in the updated PDF to clarify the modifications. In the updated version of the PDF, we have added updates to Figures 2, 5, 7, 8, 13, 17, 18, 19, and 21, as well as Tables 2 and 4, indicating the resampling and re-tessellating done for the given experiments in their respective captions.
> > >
> > > We also apologize for not properly including some of the proposed modifications in the pdf; this was an oversight, which we will correct in the final version of the paper. Since we cannot update the pdf anymore, we describe explicitly below the additional modifications that have been made to the pdf to account for the discussion above.
> > >
> > > For ``Point A`` we have modified the text of proposition 1 so that it reads:
> > >
> > > Suppose that $p,q,r$ are Lebesgue-absolutely continuous distributions on $\mathbb{R}^d$ with $p\neq q$, that $p$ and $q$ have smooth densities, and that $r$ has full support. Then, for $n_R=2$ and references points $z_1,z_2\sim r$, the probability that $\pi_*p\neq\pi_*q$ is strictly positive and hence the PQMass test is consistent as $m,n\to\infty$ with $m/n$ being bounded both above and below by strictly positive, finite scalars.
> > >
> > > Moreover, we have added the following sentence as the first line of the proof:
> > >
> > > As $m,n\to\infty$, the bounds on $m/n$ guarantee that $m$ and $n$ to grow at rates such that Fisher's test for multinomial distributions with $m$ and $n$ samples from the two distributions distinguishes them.
> > >
> > > For ``Point B``, we have added the following text immediately after the proof of proposition 1:
> > >
> > > Since $\pi$ depends on $r$, the probability referred to in Proposition 2.1 is over the choice of $n_R$ samples from $r$. Of course, for "bad" choices of reference points, we may be unlucky and not distinguish $p$ and $q$ (if the two resulting cells have the same mass under $p$ and $q$). The proposition states that, with positive probability, this does not happen.
> > >
> > > For ``Point C and D``
> > > We have replaced the last paragraph of Section 2 by the following text:
> > >
> > > Note that the $n_R\to\infty$ limit in Proposition 2.2 does not refer to performing the test with "$n_R=\infty$" to compare two $\chi^2$ distributions with infinite degrees of freedom. The proposition states that the probability that the tessellation defined using $n_R$ random reference points sampled from $r$ separates $p$ and $q$ -- *which is a well-defined quantity for any $n_R$* -- approaches 1 as $n_R\to\infty$. These results show that, at least for 'well-behaved' densities, no information is lost by the PQMass statistic. By this, we mean that if two distributions are distinct, then the test has a positive probability of distinguishing them (and the probability approaches 1 in the asymptotic limit). This contrasts with tests that approximate the distributions by a certain fixed class (e.g. tests considering only summary statistics such as the second-moment statistics).  These results can likely be generalized further, which we leave for future work.
> > >
> > > Since Proposition 2.2 is about the sensitivity of the test in the asymptotic limit, the proposition simply tells us that the sensitivity of the test approaches 1 as the number of samples grows, but does not tell us about the rate of convergence. The usefulness of the test itself is demonstrated by the empirical evidence in the experiments presented in the next section.

---

> ### Author Response · Authors · 2024-11-22
> **Response (2/4)**
>
> ### Practical implementation
>
> > As far as I can tell, you've only considered exactly balanced samples each time: m=n. While in some applications the user has control over both m and n, this will not be the case in many scenarios. So how well does the test perform when the two samples are imbalanced, including extreme cases?
>
> We have multiple experiments in which we have imbalanced datasets in Appendix F.4, in which we show PQMass performs well even when the samples are imbalanced. We have included the number of samples being used in each of the experiments for clarification. In the default implementation, PQMass draws reference points uniformly from the concatenation of the two samples, so the smaller sample is likely to have fewer references drawn from it.
>
> > Using reference samples z_i randomly taken from the two distributions (x_i) and (y_i) has a long history, even when comparing samples from two MCMC samplers (or the same MCMC sampler twice, when considering convergence diagnostics; e.g. Sisson & Fan (2007). Statistics and Computing, 17, 357-367). So this approach is fine. However, a disadvantage of this is that when p and q differ only, say, by a small mode with low weight in the tail, then by sampling uniformly over the x_i and y_j, one is reasonably likely to miss samples in this mode, as the sampling will naturally favour repeated samples in areas of higher density, and hence the test will miss the obvious difference between p and q.
>
> We should note that missing a small mode as described would affect any two-sample test by construction. One can always design a mode to be "sufficiently small" that it eludes a sample-based test as samples from it would not appear among the test samples. It is also worth noting that in the Voronoi binning step of the PQMass test, all samples are binned, such that PQMass can be sensitive to a small missing mode even without a reference sample within said mode (though it certainly helps).
>
> However, we agree with the referee that it is important to evaluate the effectiveness of PQMass at detecting a discrepant small mode, and compare it to other existing methods. To this end we have added a test using CIFAR-10 which compares the sensitivity of PQMass with other tests used for generative models to detect a small mode.
>
> We select eight classes (airplanes, automobiles, birds, cats, deer, dogs, frogs, and horses) and split the dataset into two sets $X$ and $Y$, each containing 24,000 samples (3,000 from each class). To simulate a small mode, we introduce a parameter, $\alpha$, and add $3000\cdot\alpha$ images of a new class (trucks) only to $X$, while randomly removing the same number of images of other classes to keep $X$ and $Y$ the same size. We then run various tests to compare $X$ to $Y$.
>
> In the table below, we see that as we include images from the new class, which introduces a small mode, PQMass picks up the difference bewteen $X$ and $Y$ even for small $\alpha$. On the other hand, FLD struggles to detect the difference; FID also detects the difference (as shown by the monotonically increasing test statistic), but, unlike PQMass, does not give a point of reference allowing to compute significances (p-values).
>
> | $\alpha$ | PQMass | FID | FLD
> |---------------|---------------|----------------|----------------|
> | 0.00 | 100.41 | 17.66 | -4.05 |
> | 0.01 | 100.60 | 17.77 | -4.13
> | 0.02 | 101.71 | 18.03 | -3.89
> | 0.03 | 102.71 | 18.31 | -4.16
> | 0.04 | 103.80 | 18.68 | -3.64
> | 0.05 | 104.02 | 18.96 | -4.05
> | 0.06 | 106.89 | 19.56 | -3.78
> | 0.07 | 108.00 | 20.15 | -4.15
> | 0.08 | 110.44 | 20.40 | -4.03
> | 0.09 | 112.31 | 21.11 | -4.14
> | 0.10 | 114.65 | 21.41 | -3.88
> | 0.20 | 144.75 | 27.84 | -3.78
> | 0.30 | 195.67 | 36.18 | -3.85
> | 0.40 | 261.46 | 45.21 | -3.98
> | 0.50 | 318.96 | 54.49 | -3.95
> | 0.60 | 407.73 | 64.41 | -3.69
> | 0.70 | 486.08 | 75.08 | -3.22
> | 0.80 | 579.01 | 85.00 | -3.37
> | 0.90 | 677.07 | 97.70 | -3.01
> | 1.00 | 778.88 | 108.72 | -2.90

---

> > ### Author Response · Authors · 2024-11-22
> > **Response (3/4)**
> >
> > > (The examples in Section 3 seemed to consider equally weighted mixtures only ...). Hence the need to repeat the test multiple times with different z_i. Is there any argument to be made to reconsider this equal sampling of the z_i from (x_i) and (y_j), to ensure there is a greater spread over the space, rather than doubling (and tripling, and ...) down on high density regions?
> >
> > This is a good question. Our choice of sampling references points from both the $x_i$ and $y_i$ means that we are drawing samples from the uniformly weighted mixture of the two distributions.
> >
> > Intuitively, for the test to be more discriminative, we need the regions to partition the probability masses of both distributions close to uniformly, in particular, using a finer partition in regions where *either* distribution has a high density. A tessellation obtained using references points sampled only from among the $x_i$ may not be informative about $y_i$, and conversely. Indeed, one can construct pathological cases where sampling reference points exclusively from one set gives insensitive results. This motivates our choice to sample reference points from both distributions.
> >
> > Our implementation includes multiple sampling schemes in which the user can define if they want to sample reference points from the $(x_i)$, from the $(y_j)$, or from a from a Gaussian approximating the empirical distribution of both the $x_i$ and $y_j$. The weights of the mixture $(w_P,w_Q,w_{\cal N})$ over these three distributions are an input to the test. Thus the standard form of the test corresponds to the choice $(\frac12,\frac12,0)$.
> >
> > In the table below, we consider the same experimental setup as that of Table 2 (adding progressively larger amounts of noise to one subset of ImageNet samples) and we compare five settings of PQMass, corresponding to:
> > - $(1,0,0)$ (samples from $x_i$),
> > - $(0,1,0)$ (samples from $y_j$),
> > - $(\frac12,\frac12,0)$ (samples from $x_i$ and $y_j$),
> > - $(\frac13,\frac13,\frac13)$ (samples from $x$, $y$, and an approximating Gaussian).
> > - $(0,0,1)$ (samples only from an approximating Gaussian).
> >
> > We can see that the tests tend to agree across all noise levels.
> >
> > | Noise variance | $(1,0,0)$ | $(0,1,0)$ | $(\frac12,\frac12,0)$ | $(\frac13,\frac13,\frac13)$ | $(0,0,1)$ |
> > |---------------|---------------|----------------|------------------|-----------------------------|------------------
> > | 0.00 | 99.67 | 99.30 | 99.42 | 98.44 |  97.26 |
> > | 0.01 | 105.31 | 108.03 | 105.12 | 106.45 |  107.50 |
> > | 0.02 | 120.88 | 120.94 | 120.10 | 120.57 |  126.68 |
> > | 0.03 | 140.22 | 143.19 | 141.92 | 141.98 |  150.74 |
> > | 0.04 | 164.31 | 169.39 | 164.40 | 166.63 |  176.34 |
> > | 0.05 | 186.45 | 200.02 | 188.44 | 191.24 |  207.06 |
> > | 0.06 | 213.71 | 237.17 | 219.69 | 228.54 |  238.99 |
> > | 0.07 | 231.76 | 256.84 | 247.05 | 254.90 |  271.32 |
> > | 0.08 | 279.40 | 320.69 | 291.21 | 294.87 |  304.93 |
> > | 0.09 | 311.19 | 367.82 | 325.41 | 333.71 |  341.83 |
> > | 0.10 | 338.20 | 400.67 | 354.46 | 361.37 |  373.30 |
> > | 0.15 | 489.99 | 608.69 | 693.55 | 865.16 |  799.72 |
> > | 0.20 | 782.66 | 1133.08 | 1063.95| 1278.82 |  1147.42 |
> > | 0.50 | 1453.59 | 2441.06 | 2148.51 | 2801.73 | 2421.71 |
> > | 1.00 | 2018.23 | 3564.49 | 2888.58 | 3824.30  | 3284.68 |
> >
> >
> > > Figure 2: presumably when "this process" is repeated 500 (or 200?) times, this also means resampling the z_i? (Could be clearer in the caption.) But really, this histogram doesn't really seem to follow the red density. Suggest that the number of reps is increased here considerably if you're wanting to claim these distributions are the same. Also the y-axis label of "Frequency" seems to be incorrect, as the axis ticks aren't measured in counts.
> >
> > Indeed, in the original figure the match between the PQMass values and the ideal $\chi^2$ distribution was not perfect, simply due to limited number of samples used. We have now redone this figure, showing how PQMass converges to the ideal distribution as the number of samples is increased. For this test, we have both resampled reference points, and with each retessellation used new samples from the distribution. In practical applications, the number of samples available from real data might be limited, and in such cases, the PQMass distribution can be approximated by reusing the samples and only retessellating, which also yields good results. This is what was done in some examples where the total number of samples available was limited. This shows that for typical sample sizes the match is very close and more than adequate for testing the null hypothesis.

---

> > > ### Author Response · Authors · 2024-11-22
> > > **Response (4/4)**
> > >
> > > > In the results in general, the empirical mean of the PQMass statistic (4) seems to be taken as "the statistic" for determining whether p=q (or p!=q). Is it obvious that this mean statistic has a chi-squared(n_R) distribution under the null? Each of the components of this statistic are chi-squared(n_R) (conditional on z_i) under the null, but is it obvious that they are independent from eachother? (chi-squared(m) + chi-squared(n) = chi-squared(m+n) if the two LHS distributions are independent). Certainly the alternative distributions are not the same for different z_i. So can this "mean PQMass statistic" actually be compared to the chi-squared null distribution in any meaningful way? Or is the use of the mean PQMass statistic merely limited to relative value comparisons? In which case, the paper should be clearer about all of this and tone down the chi-squared relevance.
> > >
> > > In the cases where reusing the samples between retessellation, indeed the retessellations are not independent since the samples are shared between each test. It is still possible to get a more exact test from the retessellations as described in appendix A and done in the new version of Figure 2. However, this is very computationally expensive and could be unneccessary for typical applications, as we have empirically observed that with sharing samples between retessellations the $\chi^2$ distribution is well approximated. The cases where this has been done have been made clearer in the caption of the figures and tables of the relevant experiments.
> > >
> > > Note that, in practice, while we mainly quoted the mean and standard deviation of the obtained $\chi^2_{PQM}$ distribution in the paper for the sake of making the figures concise and intuitively interpretable within limited page space when the $\chi^2$ distribution was obtained, the entire shape of the distribution yields useful information about the possibility of ruling out the null hypothesis. In cases where accuracy is paramount and retessellation without resampling was used but didn't rule out the null hypothesis, the entire shape of the distribution with resampling can always be obtained by re-running the test with resampling.
> > >
> > > > Section 4: Limitations. The method also appears to require independent samples (x_i), (y_i), otherwise the multinomial model does not hold. How likely is this in practice? In the MCMC comparisons in Table 1? Etc.
> > >
> > > The method does indeed require independent samples. This is an important point that we had not highlighted in the original version, something we have now amended. While most generative models (diffusion models, GANs, etc.) do generate independent samples, MCMC does not when reusing samples from the same chain. While in the experiment in the paper we subsampled the chain at intervals where autocorrelation was measured to be low, the samples are not truly independent. We have added a clarification about this to the experiment, and we thank the reviewer for bringing this important point to our attention.
> > >
> > > ### Other
> > >
> > > Thank you for pointing out typos, these have now been corrected.

---

> > > > ### Comment · Reviewer_6iXh · 2024-11-29
> > > >
> > > > Deviation of the statistic away from chi-squared. Thank-you for this discussion. I think it would be very useful to have a short discussion about this in the main paper, otherwise the less diligent reader will miss this important point and think that they have a perfect chi-squared test, when in fact they do not. Having this discussion doesn't weaken your method or contribution, but rather gives assurance as to when and how it can be used and interpreted, which is highly valuable.

---

> > > > > ### Author Response · Authors · 2024-12-03
> > > > >
> > > > > We agree a short discussion would be useful to clarify the point of mean-of-retessellations for the reader. We have written an appendix section to address this in more detail, which we will add to the updated draft after acceptance (as it is now not possible to update the draft). Here is the text to be included:
> > > > >
> > > > > In some of our experiments, mainly in cases where samples are expensive to generate, we have used as a metric the mean of PQMass run with many retessellations, though with the same input data. In principle, these are not independent tests and so should not produce a $\chi^2$ distribution with a mean equal to $n_r-1$. In practice, we find that, in particular, in high dimensional problems, the test run with retessellations very closely approximates a $\chi^2$ distribution (in the null case), though this is not the case in low dimensions.
> > > > >
> > > > > However, for a more robust test using retessellations, one can perform permutations of the input samples to determine a valid range for the mean-of-retessellations measurement.
> > > > >
> > > > > In this permutation test, one first computes the mean-of-retessellations like normal "test statistic," then for $n_{permute}$ iterations, one shuffles the input samples randomly and reruns the mean-of-retessellations measurement, creating a "permute statistic" histogram.
> > > > >
> > > > > The permutation tests satisfy the null hypothesis by construction since they are a mixture of the two inputs. Under the null hypothesis, the "test statistic" must be uniformly sampled from the "permute statistic" since all the samples from both inputs are iid. This can concretely determine a p-value based on the mean-of-retessellations.
> > > > >
> > > > > We showcase the "permute statistic" with the following experiment. We define two 100-dimension standard Guassians and use 1000 samples for each input. We then perform the permutation of the samples from the two standard Gaussian, and we see that the null test result has a p-value of 0.58. We then repeat the experiment, but to make the two sets of samples out-of-distribution, we scale the second standard Gaussian by 1.1. We find that the "permute statistic" returns a p-value of 0.02 for the out-of-distribution test, even though the mean-of-retesselations is within 1-$\sigma$ of the broader $\chi^2(n_r-1)$ distribution.
> > > > >
> > > > > This test removes some information from the shape of the distribution (one could perform a similar test with std-of-retessellations) and so it is still advised to look at the $\chi^2_{PQM}$ distribution for irregularities. Furthermore, this requires $n_{permute}$ times the computational resources to complete the test, which can become very large in real scenarios.
> > > > >
> > > > > Ultimately, we find PQMass is sensitive enough that, in most cases, the 'permute test' is unnecessary, and the rejection of the null hypothesis can be conservatively done based on the $\chi^2(n_r-1)$ distribution.
> > > > >
> > > > > We hope this is sufficient to address your question and to make it clearer for future curious readers.

---

> > > ### Comment · Reviewer_6iXh · 2024-11-29
> > >
> > > Mixture over (x,y,Gaussian): great that this is in your implementation. Flexibility is key - while the results are fairly consistent here across settings, it won't take long for someone to find a highly sensitive case.
> > >
> > > New Fig 2: Excellent. I am much reassured.

---

> > > > ### Author Response · Authors · 2024-12-03
> > > >
> > > > We thank the reviewer for their positive comment.

---

> > ### Comment · Reviewer_6iXh · 2024-11-29
> >
> > Reference samples with small nodes: Thank you for this study. Yes, this will affect all methods, but it's important that you demonstrate sensitivity to this choice.

---

> > > ### Author Response · Authors · 2024-12-03
> > >
> > > We thank the reviewer for their comment. We agree that this is an important test to show, and we will add it to the final version of the paper.

---

### Official Review · Reviewer_YLXe · 2024-11-05

**Soundness:** 3
**Presentation:** 3
**Contribution:** 3
**Rating:** 6
**Confidence:** 3

**Summary:**

The paper introduces PQMass, a likelihood-free approach for assessing the quality of generative models by comparing sample distributions without estimating probability densities. PQMass operates by dividing the sample space into non-overlapping regions, using chi-squared tests to measure the similarity between the distributions of real and generated samples in these regions. The method yields a p-value indicating whether samples from two sets arise from the same distribution. Authors argue that PQMass offers several advantages: it does not require auxiliary models or feature extraction, works across various data types, and scales to moderately high-dimensional data. Through experiments, authors aim to show that PQMass is effective in evaluating fidelity, diversity, and novelty in generated samples, providing a robust alternative for assessing generative models in machine learning.

**Strengths:**

- PQMass introduces a new, likelihood-free approach for evaluating generative models by comparing sample distributions without density estimation or feature extraction, which is a fresh alternative to traditional metrics like FID and MMD.

- The experiments are comprehensive, testing PQMass across various generative models, data types, and dimensions. The paper includes comparisons with established metrics, scalability tests, and ablation studies, which aims to show PQMass’s robustness, scalability, and ability to assess fidelity, diversity, and novelty effectively.

- PQMass offers a potentially practical, scalable solution for evaluating fidelity, diversity, and novelty in generative models across diverse, high-dimensional data domains, filling relevant gaps in current model evaluation techniques.

**Weaknesses:**

- The use of L2 or L1 distance metrics may limit PQMass's effectiveness, especially when the data resides on a complex manifold. These metrics may not capture meaningful differences in such cases, potentially leading to inaccurate assessments.

- Experimental evaluation mostly focus on synthetic data or standard datasets like MNIST and CIFAR-10. Testing on more complex, real-world datasets would strengthen PQMass’s claims about the performance of the proposal.

- Although the paper briefly addresses other data types, it primarily evaluates PQMass on image data. Expanding experiments to more complex, structured data (e.g., text or biological data) and analyzing the impact of different distance metrics in these contexts would help to better validate PQMass performance.

**Questions:**

I would like to hear authors' opinions about the detailed weaknesses of their proposal.

---

> ### Author Response · Authors · 2024-11-22
> **Response**
>
> We thank the reviewer for their very helpful comments. We have tried to address all the issues raised to the best of our abilities.
>
> > The use of L2 or L1 distance metrics may limit PQMass's effectiveness, especially when the data resides on a complex manifold. These metrics may not capture meaningful differences in such cases, potentially leading to inaccurate assessments.
>
> This is a great suggestion. To address this, we have decided to expand the discussion of the choice of the distance metric. We have added an extra appendix (Appendix G) to discuss this. In the extra experiments presented here, we have explored the performance of PQMass using 8 different common distance metrics. All metrics perform as expected, resulting in a $\chi^2$ distribution for the null test (Figure 20 and Figure 21, left panel) but show different amounts of sensitivity for differentiating two distinct distributions, as you pointed out. We also repeat the experiment in Section 3.3 -- adding noise to ImageNet -- with the same eight metrics. In Figure 20 we show that all metrics correctly result in the $\chi^2$ distribution for the null test. We also show the metrics as noise of different variance is added and showcase that the choice of metric will have an effect on results.
>
> Note that although a suboptimal choice of distance metric could lead to a less informative statistic (in the sense that the test could potentially not differentiate two distributions with confidence), the test will not falsely rule out the null hypothesis. Our general suggestion is to use the Euclidean metric by default, but in cases where this metric is not sufficiently informative one could choose another appropriate metric, such as one defined in feature space.
>
> > Experimental evaluation mostly focus on synthetic data or standard datasets like MNIST and CIFAR-10. Testing on more complex, real-world datasets would strengthen PQMass’s claims about the performance of the proposal.
>
> We also felt that standard datasets were not enough to test this method (though we would like to point out here that many competing methods were only tested on these benchmarks when introduced). This is why we included tests on higher-dimensional benchmarks like ImageNet and presented experiments on real-world complex datasets from the field of astrophysics and tabular census data. These can be found in Appendices F.3 and F.4, and we are happy to point to them more prominently in the main text.
>
> > Although the paper briefly addresses other data types, it primarily evaluates PQMass on image data. Expanding experiments to more complex, structured data (e.g., text or biological data) and analyzing the impact of different distance metrics in these contexts would help to better validate PQMass performance.
>
> Again, other data types were added to the appendix due to space limitations (see Appendices F.2 and F.3). In addition, and in response to the referee's helpful suggestions, we have added a new experiment on protein sequences to Appendix F.5. We believe that this experiment addresses all three weaknesses addressed by the referee: It does not use the L2 distance metric, it is a complex real-world dataset, and uses complex, structured data.
>
> > I would like to hear authors' opinions about the detailed weaknesses of their proposal.
>
> We tried our best to address the weaknesses we found in the limitations section (sec. 4). However, if the reviewer feels like there are any missing limitations in that section, we would be happy to add them.

---

> > ### Comment · Reviewer_YLXe · 2024-11-25
> >
> > Thanks for your thorough response. I've been reading as well the other reviews and discussions. I clearly lean towards acceptance for this paper, but I prefer to wait until the discussion with other reviewers before increasing my score.

---

> > > ### Author Response · Authors · 2024-12-03
> > >
> > > We thank the reviewer for their engagement with our work during the referee process, and we do believe that the quality of our paper has been greatly improved thanks to their comments. As the deadline for updates is approaching, we would like to remind the reviewer to update their final score as they deem appropriate.

---

### Meta-Review · Area_Chair_hKnv · 2024-12-23

**Metareview:**

This paper suggests a method to assess how good a generative model is by a likelihood free probability mass estimation procedure. While I find some aspects heuristic, the reviewers are in agreement that there are good ideas here and their opinions improved after the rebuttal period. The method is appealing in its lean reliance on assumptions and its scalability.

**Additional Comments On Reviewer Discussion:**

reviewer discussion was helpful here

---

### Decision · Program_Chairs · 2025-01-22

Accept (Poster)